# Rethinking Confidence Scores and Thresholds in Pseudolabeling-based SSL

**Harit Vishwakarma** [* 1]  **Yi Chen** [* 2]  **Satya Sai Srinath Namburi GNVV** [3]  **Sui Jiet Tay** [4]
**Ramya Korlakai Vinayak** [2]  **Frederic Sala** [1]

## Abstract

Modern semi-supervised learning (SSL) methods rely on pseudolabeling and consistency regularization. Pseudolabeling is typically performed by comparing the model's confidence scores and a predefined threshold. While several heuristics have been proposed to improve threshold selection, the underlying issues of overconfidence and miscalibration in confidence scores remain largely unaddressed, leading to inaccurate pseudolabels, degraded test accuracy, and prolonged training. We take a first-principles approach to learn confidence scores and thresholds with an explicit knob for error. This flexible framework addresses the fundamental question of optimal scores and threshold selection in pseudolabeling. Moreover, it gives practitioners a principled way to control the quality and quantity of pseudolabels. Such control is vital in SSL, where balancing pseudolabel quality and quantity directly affects model performance and training efficiency. Our experiments show that, by integrating this framework with modern SSL methods, we achieve significant improvements in accuracy and training efficiency. In addition, we provide novel insights on the trade-offs between the choices of the error parameter and the end model's performance.

## 1. Introduction

The lack of high-quality labeled data is a major bottleneck in training high-accuracy models. The semi-supervised learning (SSL) paradigm tackles this problem by leveraging abundant unlabeled data alongside a limited set of labeled examples (Chapelle et al., 2006; Zhu, 2005; van Engelen

---

[*]Equal contribution [1]Department of Computer Sciences, University of Wisconsin-Madison, WI, USA [2]Department of Electrical and Computer Engineering, University of Wisconsin-Madison, WI, USA [3]GE HealthCare [4]NYU Courant Institute. Correspondence to: Harit Vishwakarma <hvishwakarma@cs.wisc.edu>, Yi Chen <yi.chen@wisc.edu>.

*Proceedings of the 42ⁿᵈ International Conference on Machine Learning*, Vancouver, Canada. PMLR 267, 2025. Copyright 2025 by the author(s).

& Hoos, 2019). While SSL dates back decades and includes a wide variety of approaches, modern SSL methods frequently rely on a pair of ideas: *self-training or pseudolabeling* – where model generated labels are assigned to unlabeled data for further training (McLachlan, 1975; Amini et al., 2023; Rosenberg et al., 2005; Lee, 2013; Rizve et al., 2021) – and *consistency regularization* to enforce stability in predictions across perturbed inputs (Laine & Aila, 2017; Bachman et al., 2014; Sajjadi et al., 2016; Fan et al., 2021; Kukačka et al., 2017). SSL techniques with these ideas have strongly performed on several benchmark datasets.

While pseudolabeling is a powerful technique, its effectiveness hinges on a fundamental question: *which points should be labeled using the model's predictions?* Since pseudolabels are derived from a model being trained, they can be highly unreliable. A naive approach that assigns pseudolabels too liberally risks injecting noisy labels into training, amplifying the model's existing errors – a phenomenon known as confirmation bias. Conversely, an overly conservative approach that selects only the correct predictions severely limits the amount of useful training data. Both extremes can degrade SSL performance, leading to either poor generalization or slow convergence. To fully harness the potential of pseudolabeling, we need a principled approach for pseudolabeling with explicit control on these trade-offs.

A widely used strategy pseudolabels points on which the model's confidence score exceeds a threshold. This approach provides a simple mechanism for selecting points while controlling the quality and quantity of pseudolabels via thresholds. However, the prior works based on this approach suffer from two key limitations that restrict their effectiveness. First, thresholding techniques are often heuristic-driven, lacking precise control for a target error level (Sohn et al., 2020; Zhang et al., 2021; Wang et al., 2023; Xu et al., 2021; Chen et al., 2023; Li et al., 2023). Second, commonly used scores, such as the model's softmax outputs, tend to be unreliable. Recent studies in this vein (Loh et al., 2022; Mishra et al., 2024; Rizve et al., 2021) have highlighted issues of overconfidence and miscalibration in these scores, leading to inefficiencies in pseudolabeling.

In addition to these problems with the choices of scores and thresholds, an equally important question remains: how

much error should be allowed in pseudolabeling? As discussed earlier, a very low tolerance may pseudolabel too few points, and conversely, a high tolerance may allow for large errors, hurting the efficiency of pseudolabeling in both cases. To solve these issues, we seek *a principled solution to get confidence scores and thresholds for pseudolabeling at a given target level of error tolerance.*

We take a first principles approach to formalize the pseudolabeling objective: maximize the number of pseudolabeled points while adhering to the given error tolerance. We formulate this objective as an optimization problem over a flexible space of confidence functions and thresholds. By solving this optimization problem, we obtain confidence scores tailored to our objective of pseudolabeling. To ensure the pseudolabeling error constraint is strictly followed, we use a separate procedure to estimate thresholds on these scores using part of the validation data. Pseudolabeling with these scores and thresholds ensures we are pseudolabeling a maximal set of points that can be pseudolabeled at the given error tolerance level.

We integrate this approach into popular pseudolabeling-based SSL methods, providing two benefits. First, it provides a principled way to derive confidence scores and thresholds for any given error tolerance. Second, by enabling more precise pseudolabeling, it improves the utilization of the unlabeled data and is expected to yield an end model with higher test accuracy compared to the ad-hoc choices of scores and thresholds.

Our main contributions are summarized as follows,

1. Our work settles the question of the right choices of confidence scores and thresholds in pseudolabeling-based SSL methods by introducing a framework for learning confidence scores and thresholds. Departing from heuristic-driven or ad-hoc and unreliable choices of scores and thresholds, this framework provides principled choices of scores and thresholds for pseudolabeling with any target error tolerance.

2. We show how this framework for learning confidence scores and thresholds can work in concert with popular SSL methods such as Fixmatch (Sohn et al., 2020), Freematch (Wang et al., 2023), etc., and conduct an extensive empirical evaluation demonstrating that by pseudolabeling using confidence scores and thresholds learned from our method can yield significant improvements in the test accuracy.

3. Leveraging our framework's ability to pseudolabel at any target error level, we study the impact of varying pseudolabeling error levels—from fixed to dynamic tolerance throughout training. Our results confirm the intuition that lower pseudolabeling errors lead to better end models compared to higher errors. Moreover,

among dynamic schedules, it is better to use a decreasing schedule of error tolerances.

## 2. Related Work

**Semi-supervised learning (SSL).** There is a rich literature on SSL (Zhu, 2005; Chapelle et al., 2006; Singh et al., 2008; Oliver et al., 2018). This literature comprises of a wide variety of approaches. Among these, significant focus has been placed on self-training (also called pseudolabeling) (Scudder, 1965; Blum & Mitchell, 1998; Rosenberg et al., 2005; Lee, 2013; Oymak & Gulcu, 2020; Amini et al., 2023), generative models (Nigam et al., 2000; Adams & Ghahramani, 2009; Kingma et al., 2014), graph-based strategies (Blum & Chawla, 2001; Niyogi, 2013; Subramanya & Talukdar, 2022), and transductive approaches (Vapnik et al., 1998; Joachims, 1999). Due to their simplicity, pseudolabeling-based approaches have gained prominence and are widely used in application areas such as NLP (Karamanolakis et al., 2021), speech recognition (Kahn et al., 2020), and protein prediction (El-Manzalawy et al., 2016). Our paper focuses on recent variants of this, discussed next.

**Pseudolabeling based SSL.** These methods generate artificial labels for unlabeled data and use them for training the model. A crucial challenge here is the issue of confirmation bias (Arazo et al., 2020), i.e., when a model starts to reinforce its own mistakes. To overcome this and to maintain a high quality of pseudolabels, confidence-based thresholding is applied. Here, only the unlabeled data with confidence higher than a particular threshold is used (Sohn et al., 2020). Due to the limitations of fixed thresholds, adaptive thresholds based on the classifier's learning status have been introduced to improve performance (Xu et al., 2021; Zhang et al., 2021; Wang et al., 2023). Nearly all of these methods also use some form of consistency regularization (Laine & Aila, 2017; Bachman et al., 2014; Sajjadi et al., 2016; Fan et al., 2021; Kukačka et al., 2017) where the core idea is that the model should produce similar prediction when presented with different versions (perturbations) of inputs and all the present SSL methods (Xie et al., 2020; Wang et al., 2023; Sohn et al., 2020; Zhang et al., 2021; Chen et al., 2023; Zheng et al., 2022; Xu et al., 2021; Roelofs et al., 2022).

**Confidence functions and calibration.** Miscalibration (overconfidence) in neural networks plagues various applications (Nguyen et al., 2015; Hendrycks & Gimpel, 2017; Guo et al., 2017), including SSL. To mitigate this in general, a range of solutions have been proposed, including training-time methods (Moon et al., 2020; Kumar et al., 2018; Hui et al., 2023; Corbière et al., 2019; Foret et al., 2021) and post-hoc methods (Guo et al., 2017; Kumar et al., 2019; Gupta & Ramdas, 2022; Kull et al., 2019; Zadrozny & Elkan, 2002). In pseudolabeling based SSL, recent works (Rizve et al., 2021; Loh et al., 2023; Mishra et al., 2024)

noted the issue of miscalibration. To promote calibration, Loh et al. (2023) use Bayesian neural nets by replacing the model's final layer with a Bayesian layer. Rizve et al. (2021) utilize negative labels and an uncertainty-aware pseudolabel selection technique. Mishra et al. (2024) incorporate a regularizer to encourage calibration.

While calibration is a reasonable goal in general, it may not be sufficient to address the overconfidence problem in SSL and other applications. In pseudolabeling, we seek the use of scores that can easily segregate the model's correct and incorrect predictions, which is closely related to the ordinal ranking criterion (Hendrycks & Gimpel, 2017; Moon et al., 2020; Foret et al., 2021; Corbière et al., 2019). Rather than experimenting with several such choices, ideally, we would have a flexible framework that can learn confidence functions explicitly optimizing pseudolabeling objectives.

## 3. Background and Problem Setup

We begin with notation, then provide useful background and a statement of our goal.

**Notation.** Consider a feature space $\mathcal{X}$ and label space $\mathcal{Y} = \{1, \ldots, k\}$ in a $k$-class classification task. As usual in semi-supervised learning, we have access to a set $X_u = \{\mathbf{x}_u\}_{u=1}^{n_u}$ of unlabeled data drawn from the distribution $P_x$ over $\mathcal{X}$. We also have access to $D_l = \{(\mathbf{x}_l, y_l)\}_{l=1}^{N_l}$, a set of labeled data points drawn from the joint distribution $P_{xy}$, with $N_l \ll n_u$. Let $h : \mathcal{X} \to \mathcal{Y}$ denote a model and $g : \mathcal{X} \to T^k \subseteq \mathbb{R}^k$ be an associated confidence function giving a score $g(\mathbf{x})$ indicating the confidence of $h$ on its prediction for any data point $\mathbf{x}$. For any $\mathbf{x}$ the hard label prediction is $\widehat{y} := h(\mathbf{x})$. When the prediction $\widehat{y}$ is used as a pseudolabel, we denote it as $\tilde{y}$. In general, for a vector $\mathbf{v} \in \mathbb{R}^d$, $\mathbf{v}[i]$ denotes its $i-$th component. The vector $\mathbf{t}$ denotes thresholds over the scores $k$-classes, and $\mathbf{t}[y]$ is its $y-$th entry, i.e., the threshold for class $y$.

### 3.1. Pseudolabeling-based Semi-Supervised Learning

Given, as above, a large collection of unlabeled data $X_u$ and a small set of labeled points $D_l$, inductive semi-supervised learning (SSL) seeks to learn a classifier $\widehat{h}_{\text{ssl}}$ from the model class $\mathcal{H}$. The promise of SSL is that by effectively using $X_u$ in the learning process, it can learn a better classifier than its supervised counterpart, which learns only from $D_l$.

In many recent pseudolabeling-based SSL techniques, in each iteration of training, a batch of labeled and unlabeled data is obtained, then the sum of the losses $\widehat{\mathcal{L}} = \widehat{\mathcal{L}}_s + \lambda_u \widehat{\mathcal{L}}_u + \lambda_r \widehat{\mathcal{L}}_r$ is minimized with respect to the model $h$. Here $\widehat{\mathcal{L}}_s$ is the supervised loss, $\widehat{\mathcal{L}}_u$ unsupervised loss, and $\widehat{\mathcal{L}}_r$ is (the sum of) regularization term(s). The positive constants $\lambda_u, \lambda_r$ are hyperparameters controlling the relative importance of the corresponding terms.

**Supervised loss.** Given a batch of labeled data, $D_l^b$, the supervised loss is computed as follows, $\widehat{\mathcal{L}}_s(h \mid D_l^b) = \frac{1}{|D_l^b|} \sum_{(\mathbf{x},y) \in D_l^b} H(y, h, \mathbf{x})$. Here $H(y, h, \mathbf{x})$ is the standard cross-entropy loss between the 1-hot representation of $y$ and the softmax output of $h$ on input $\mathbf{x}$.

**Unsupervised loss and consistency regularization.** For the unlabeled batch $X_u^b$, pseudolabels $\tilde{y} = h(\mathbf{x})$ are computed for each $\mathbf{x} \in X_u^b$. Then, a pseudolabeling mask, $S(\mathbf{x}, g, \mathbf{t} \mid h) = \mathbb{1}(g(\mathbf{x})[\tilde{y}] \geq \mathbf{t}[\tilde{y}])$ is computed. It is 1 for points having confidence score $g(\mathbf{x})[\tilde{y}]$, bigger than pre-determined threshold $\mathbf{t}[\tilde{y}]$, corresponding to the predicted class $\tilde{y}$. Recent methods, couple this loss and consistency regularization together by doing pseudolabeling on weakly augmented data using weak transform $\omega : \mathcal{X} \mapsto \mathcal{X}$ and then defining the cross-entropy loss on the strongly augmented data using strong transformation $\Omega : \mathcal{X} \mapsto \mathcal{X}$. The loss is,

$$\widehat{\mathcal{L}}_u := \frac{1}{|\widetilde{D}_u^b|} \sum_{(x,\tilde{y}) \in \tilde{D}_u^b} S(\omega(\mathbf{x}), g, \mathbf{t} | h) \cdot H(\tilde{y}, h, \Omega(\mathbf{x})).$$

**Regularization.** A regularization term (or a sum of multiple regularizers) is often included along with the above two losses to encourage desired behavior(s). For instance, Freematch (Wang et al., 2023) adds a self-adaptive class fairness regularizer to encourage diverse predictions during the initial training phase. Similarly, a regularizer is introduced in (Mishra et al., 2024) to encourage calibration in the model's confidence scores. Including such regularizers has been fruitful in pseudolabeling-based SSL.

### 3.2. Quality and Quantity of Pseudolabels

Given a classifier $h$, the quality and quantity of the pseudolabels w.r.t. to score function $g$ and thresholds $\mathbf{t}$, are:

**Pseudolabeling coverage (quantity).** Given a set of points $X$, the pseudolabeling coverage is the fraction of points that are pseudolabeled using $h, g$ and $\mathbf{t}$. This measurement captures the quantity of pseudolabels and is defined as

$$\widehat{\mathcal{P}}(g, \mathbf{t} \mid h, X) := \frac{1}{|X|} \sum_{(\mathbf{x}) \in X} S(\mathbf{x}, g, \mathbf{t} \mid h) \quad (1)$$

$$\mathcal{P}(g, \mathbf{t} \mid h) := \mathbb{E}_{\mathbf{x}}[S(\mathbf{x}, g, \mathbf{t} \mid h)]. \quad (2)$$

**Pseudolabeling error (quality).** This is the fraction of pseudolabeled points that got incorrect labels. This metric captures the quality of pseudolabels:

$$\widehat{\mathcal{E}}(g, \mathbf{t} \mid h, D) := \frac{\sum_{(\mathbf{x},y,\tilde{y}) \in D} S(\mathbf{x}, g, \mathbf{t} \mid h) \cdot \mathbb{1}(h(\mathbf{x}) \neq y)}{\sum_{(\mathbf{x},y,\tilde{y}) \in D} S(\mathbf{x}, g, \mathbf{t} \mid h)}, \quad (3)$$

$$\mathcal{E}(g, \mathbf{t} \mid h) = \frac{\mathbb{E}_{\mathbf{x}}[S(\mathbf{x}, g, \mathbf{t} \mid h) \cdot \mathbb{1}(h(\mathbf{x}) \neq y)]}{\mathcal{P}(g, \mathbf{t}|h)}. \quad (4)$$

## 3.3. Our Goals

Pseudolabeling-based SSL aims to learn a classifier $\widehat{h}_{\text{ssl}}$ that generalizes well on the unseen data, i.e., has high test accuracy. This is typically achieved by pseudolabeling points using confidence scores and thresholds and incorporating them into training. However, existing choices of scores and thresholding strategies are often ad hoc and unreliable, limiting their effectiveness. Departing from these unreliable approaches, our goal is to:

(i) Design principled solutions for confidence scores and thresholding to maximize the number of pseudolabeled points while ensuring the pseudolabeling error is at most $\epsilon$.

(ii) Incorporate these in the existing pseudolabeling-based SSL methods and assess whether this gives a better end model $\widehat{h}_{\text{ssl}}$.

(iii) Study the sensitivity of the SSL pipeline to pseudolabeling errors by leveraging the ability of our approach to explicitly ensure the pseudolabeling error remains below $\epsilon$.

## 4. Methodology

In this section, we discuss our principled framework to learn scores and thresholds with explicit control of the pseudolabeling errors and use them in pseudolabeling-based SSL. This framework is an adaptation of a recent solution designed to improve confidence functions for auto-labeling (Vishwakarma et al., 2024). The detailed steps are outlined in Algorithm 3 in the Appendix.

### 4.1. Pseudolabeling Optimization Framework

Given the current model $\widehat{h}_i$ in the $i$th iteration, can we obtain confidence scores and thresholds using which we can identify a maximal set of points that can be pseudolabeled using $\widehat{h}_i$ with at most $\epsilon$ error? We begin with a theoretical formulation to learn such scores and thresholds, and then introduce its practical version.

**Theoretical framework.** Instead of improving calibration or heuristics for thresholding, we propose to express the objective of pseudolabeling as an optimization problem over the space of confidence functions and thresholds. The objective is to maximize the quantity, i.e., the pseudolabeling coverage (eq. (2)) while keeping the pseudolabeling error (eq. (4)) below a tolerance level $\epsilon \in (0, 1)$. More specifically, given the classifier $\widehat{h}_i$ in any iteration $i$ of SSL,

$$g_i^{\star}, \mathbf{t}_i^{*} \in \underset{g \in \mathcal{G}, \mathbf{t} \in T^k}{\arg\max} \; \mathcal{P}(g, \mathbf{t} \mid \widehat{h}_i) \text{ s.t. } \mathcal{E}(g, \mathbf{t} \mid \widehat{h}_i) \leq \epsilon, \quad (5)$$

are the optimal confidence functions and thresholds for pseudolabeling using $\widehat{h}_i$'s predictions such that the pseudolabeling error is bounded by $\epsilon$. This frees us from arbitrary choices of confidence scores, calibration techniques, and

thresholding heuristics. Instead, solving the optimization problem over a flexible enough space will subsume specific strategies. Next, we discuss how to make this framework tractable to obtain scores and thresholds in practice.

**Practical version.** The optimization problem discussed earlier involves population-level quantities which are usually not accessible in practice. Thus we have to use their finite sample estimates and smooth variations to make the optimization problem tractable. Specifically, the coverage and error are estimated using a small amount of held-out labeled data (called calibration data $D_{\text{cal}}$) curated from the validation data. Then differentiable surrogates for the 0-1 variables are introduced. Let $\sigma(\alpha, z) := 1/(1 + \exp(-\alpha z))$ denote the sigmoid function on $\mathbb{R}$ with scale parameter $\alpha \in \mathbb{R}$. The surrogates are as follows,

$$\widetilde{\mathcal{P}}(g, \mathbf{t} \mid h, D_{\text{cal}}) := \frac{1}{|D_{\text{cal}}|} \sum_{(\mathbf{x}, y, \widetilde{y}) \in D_{\text{cal}}} \sigma\big(\alpha, g(\mathbf{x})[\widetilde{y}] - \mathbf{t}[\widetilde{y}]\big), \tag{6}$$

$$\widetilde{\mathcal{E}}(g, \mathbf{t} \mid h, D_{\text{cal}}) := \frac{\sum_{(\mathbf{x}, y, \widetilde{y}) \in D_{\text{cal}}} \mathbb{1}\big(y \neq \widetilde{y}\big) \, \sigma\big(\alpha, g(\mathbf{x})[\widetilde{y}] - \mathbf{t}[\widetilde{y}]\big)}{\sum_{(\mathbf{x}, y, \widetilde{y}) \in D_{\text{cal}}} \sigma\big(\alpha, g(\mathbf{x})[\widetilde{y}] - \mathbf{t}[\widetilde{y}]\big)} \tag{7}$$

Using these surrogates, the following practical optimization problem is obtained. It is also converted into unconstrained formulation by introducing the penalty term $\lambda \in \mathbb{R}^{+}$ controlling the relative importance of the error and coverage.

$$\widehat{g}_i, \widehat{\mathbf{t}}_i' \in \underset{g \in \mathcal{G}, \mathbf{t} \in T^k}{\arg\min} \; -\widetilde{\mathcal{P}}(g, \mathbf{t} \mid \widehat{h}_i, D_{\text{cal}}) + \lambda \, \widetilde{\mathcal{E}}(g, \mathbf{t} \mid \widehat{h}_i, D_{\text{cal}}) \tag{P1}$$

We use 2-layer neural nets as a choice of $\mathcal{G}$ and $T^k = [0, 1]^k$. The optimization problem (P1) is non-convex but differentiable and we solve it using Stochastic Gradient Descent (SGD). See Appendix C for more details on our choice of $\mathcal{G}$ and training details and hyperparameters.

### 4.2. Threshold Estimation

While we can obtain both the confidence scores and thresholds by solving (P1), we propose to estimate thresholds separately on a held-out part of the validation data to avoid potential generalization issues due to learning them simultaneously from the same data $D_{\text{cal}}$ and ensure that the pseudolabeling error constraint is strictly adhered to.

When dealing with datasets containing many classes there may not be enough samples per class to estimate reliable thresholds. Thus, to accommodate these possibilities we consider two variations of the threshold estimation procedure, (i) estimate a common (joint) threshold for all classes and (ii) estimate separate (classwise) thresholds for each class. The procedures are outlined in Algorithm 2 and Algorithm 1 in the Appendix. We discuss them briefly here.

The procedure takes in a confidence function $\widehat{g}_i$ and part of the held-out validation data referred to as $D_{\text{th}}$. The idea is

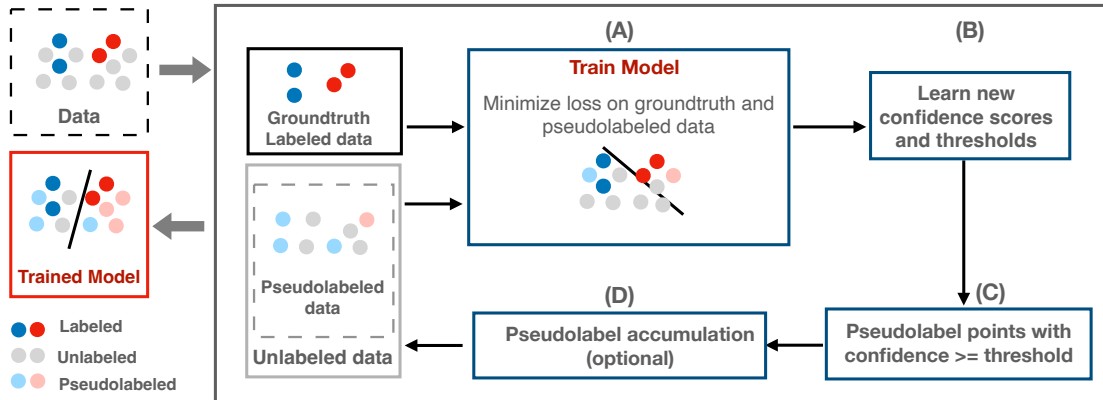

*Figure 1.* Workflow of pseudolabeling-based SSL with `PabLO` (A): Train model with standard supervised loss, consistency regularization, and other regularizers (B): Learn new confidence scores and thresholds (C): Pseudolabel points with scores greater than the estimated thresholds (D): An optional pseudolabeling accumulation to use previous pseudolabels for points that are not pseudolabeled in current round. Note, that the pseudolabels can be noisy (incorrect). The training and pseudolabeling loop continues until a pre-specified number of iterations. In the end, it outputs a model $\widehat{h}_{\mathrm{ssl}}$ that is expected to have higher test accuracy than the model trained only on the given groundtruth training data. Note that in the end, there might be points left unlabeled, and the pseudolabels might be noisy.

to estimate errors at several thresholds on this data and then pick the smallest threshold. This can be done separately for each class to obtain classwise thresholds or a common threshold for all classes. We discuss classwise thresholding here. First the data $D_{\mathrm{th}}$ is partitioned into $k$ subsets $D_{\mathrm{th}}^{(y)}$ corresponding to each class $y \in \mathcal{Y}$. Here, we slightly abuse notation: instead of $\mathbf{t} \in T^k$, we use $t \in T$ in the estimate of pseudolabeling error at threshold $t$ for class $y$. To obtain threshold $\widetilde{\mathbf{t}}[y]$ for class $y$, the procedure finds the smallest $t \in T$ such that $\widehat{\mathcal{E}}(\widehat{g}_i, t \mid h, D_{\mathrm{th}}^{(y)}) + C_1 \widehat{\sigma}(\widehat{\mathcal{E}}) \leq \epsilon$. Here $C_1$ is a constant (we use $C_1 = 0.25$) and $\widehat{\sigma}(z) = \sqrt{z \cdot (1 - z)}$ and $\widehat{\mathcal{E}}$ is used for brevity in place of $\widehat{\mathcal{E}}(\widetilde{g}_i, t \mid h, D_{\mathrm{th}}^{(y)})$. The same process is used for joint threshold estimation, where a single threshold $t$ is estimated using the entire $D_{\mathrm{th}}$ and the same $t$ is used for all classes. Using the thresholds found using these procedures ensures pseudolabeling error remains below (or close to) the tolerance level $\epsilon$.

*Remarks.* Departing from fixed thresholds as in (Sohn et al., 2020), prior works have proposed adaptive and class-wise heuristic thresholding schemes based on the model's learning status, such as in (Djurisic et al., 2023; Zhang et al., 2021; Wang et al., 2023) and others. In contrast, our approach is a principled way to estimate adaptive and class-wise pseudolabeling thresholds while providing strict control over the quality of pseudolabels. Similar procedures have been used in the context of creating reliable datasets and are backed by theoretical guarantees for the quality of pseudolabels produced (Vishwakarma et al., 2023).

### 4.3. Pseudolabeling and Accumulation

In the usual pseudolabeling-based SSL setups, the pseudolabels inferred by the model for a mini-batch are discarded after each iteration. Moreover, it is not guaranteed that a previously pseudolabeled point will get pseudolabled in the current iteration as well. Given the quality of pseudolabels is high, it is appealing to reuse the past pseudolabel for a point that did not get pseudolabeled in the current iteration. We propose to do so for techniques where the quality of pseudolabels is assured. We refer to this as "pseudolabel accumulation".

Mathematically, let $\widetilde{y}_j^{(i-1)} = \widetilde{Y}_u^{(i-1)}[j]$ and $\widetilde{y}_j^{(i)} = \widetilde{Y}_u^{(i)}[j]$ be the previous and current (fresh) pseudolabels for $j$th unlabeled point. Let the corresponding masks (indicating whether the score is above the threshold) for these psuedolabels be $S_u^{(i-1)}[j]$ and $S_u^{(i)}[j] = \mathbb{1}(\widehat{g}_i(\mathbf{x}_j)[\widetilde{y}_j^{(i)}] \geq \widehat{\mathbf{t}}_i[\widetilde{y}_j^{(i)}])$. Then with accumulation,

$$\widetilde{Y}_u^{(i)}[j] \leftarrow S_u^{(i)}[j]\widetilde{Y}_u^{(i)}[j] + (1 - S_u^{(i)}[j])\widetilde{Y}_u^{(i-1)}[j],$$
$$S_u^{(i)}[j] \leftarrow S_u^{(i)}[j] \vee S_u^{(i-1)}[j].$$

Here $\vee$ is the boolean `or` operation and the steps are executed in the order. In words, if the point is pseudolabeled in the current iteration (i.e., its current mask is 1), then it will use the current pseudolabel otherwise, if the point was pseudolabeled in earlier iteration(s) it will use the pseudolabel from that iteration and mark the point as pseudolabeled. In case the point is not pseudolabeled in this iteration or any other iteration in the past, it will remain unlabeled. While it is appealing to use this trick, its use is only warranted in settings ensuring high-quality pseudolabels. We try to understand the consequences of the inclusion and exclusion of this trick in pseudolabeling-based SSL via experiments discussed in the next section.

We put together the steps for learning scores, thresholds, and performing (optional) accumulation in a common template

| Dataset | Backbone Model $h$ | $k$ | $n_u$ | $N_{\text{val}}$ | $N_{\text{test}}$ | $N_l$ | $N_{\text{cal}}$ | $N_{\text{th}}$ | Augmentation |
|---------|--------------------|-----|-------|------------------|-------------------|-------|------------------|-----------------|--------------|
| CIFAR-10 | WRN-28-2 | 10 | 50K | 6K | 4K | 40 or 250 | 1K | 1K | Weak, Strong |
| CIFAR-100 | WRN-28-2 | 100 | 50K | 6K | 4K | 400 or 2500 | 3K | 3K | Weak, Strong |
| SVHN | WRN-28-2 | 10 | 604,388 | 15,620 | 10,412 | 40 or 250 | 3K | 3K | Weak, Strong |

*Table 1.* Details of the dataset we use in our experiments. $k$ is the number of classes. $N_l$ is the number of labeled data points used for training the backbone model $h$. $n_u$ is the number of unlabelled data points used for consistency regularization and pseudolabeling for all the methods. $N_{\text{val}}$ is the number of points used for model selection in all methods. $N_{\text{test}}$ is the number of test data points. $N_{\text{cal}}$ is the number of points used for learning the $g$ function. $N_{\text{th}}$ is the number of data points used for threshold estimation.

of pseudolabeling-based SSL algorithms. We refer to this adapted method (Algorithm 3 in Appendix B) as PabLO. The high-level steps are also illustrated in Figure 1. Next, we discuss the empirical evaluation of PabLO and baselines.

## 5. Experiments

We conduct empirical evaluation over several settings to,

**C1.** Verify that the adaptations of popular pseudolabeling-based SSL methods with PabLO output models with better test accuracy.

**C2.** Study the effects of choice of error tolerance $\epsilon$ on test accuracy of the final model.

**C3.** Understand the role of pseudolabel accumulation in our method and baselines.

### 5.1. Experimental Setup

First, we briefly describe the experimental setup, with details deferred to Appendix C. The code is available on GitHub [1].

**Methods.** We use two simple base methods that capture the core ideas of pseudolabeling (PL) and consistency regularization (CR). The first is *Fixmatch* (Sohn et al., 2020), which uses fixed thresholds on (maximum softmax probability) MSP scores for PL and CR. *Freematch* (Wang et al., 2023) improves upon it by using adaptive, class-wise thresholds and class fairness regularization (CFR) along with CR, and is a promising method among others using dynamic thresholds for PL. We include their combinations with recently proposed *Bayesian Model Averaging (BAM)* (Loh et al., 2023) and *Margin Regularization (MR)* [2] (Mishra et al., 2024) to improve calibration in SSL. We replace the pseudolabeling component by our method PabLO to obtain *Fixmatch + Ours* (a combination of PabLO and CR) and *Freematch + Ours* (a combination of PabLO, CR, and CFR). We provide implementations of these in the code submitted along with the paper.

**Datasets.** We experiment with three datasets: *CIFAR-10* (Krizhevsky et al., 2009) is an image dataset with 10 classes.

*CIFAR-100* (Krizhevsky et al., 2009) is an extended version of CIFAR-10 with 100 classes. *SVHN* (Netzer et al., 2011) is a 10-class image dataset of digits from Google Street View. More details are summarized in Table 1. We use a portion of the validation data ($D_{\text{val}}$) for our method, split into $D_{\text{cal}}$, used to learn the function $g$, and $D_{\text{th}}$, used to estimate the threshold. Unless otherwise mentioned, we use $N_l$ as 250 for CIFAR-10 and SVHN and 2500 for CIFAR-100 in our experiments.

**Adjusted iterations for baselines.** Empirically, our method requires more time to run compared to base SSL techniques. Therefore, we adopt the following strategy to ensure a fair comparison between the baselines and our method: First, we train our method for 25K iterations and obtain the average per iteration time, denoted as $\alpha_o$. Then, we train each baseline method $b$ for 5K iterations and obtain the average per iteration time, denoted as $\alpha_b$. Using these two values, we obtain the adjusted number of iterations, $\frac{\alpha_o}{\alpha_b} \times 25000$, for the baseline method $b$. Coincidentally, baselines under the same dataset have similar runtimes. We, therefore, set the adjusted number of iterations on a dataset level. For CIFAR-10, the adjusted number of iterations for baselines is 37,000. For CIFAR-100, the adjusted number of iterations for baselines is 70,000. For SVHN, it is 145,000.

**Models and training.** The backbone encoder is a Wide ResNet-28-2 for all the datasets. We use the default hyperparameters and dataset-specific settings (learning rates, batch size, optimizers, and schedulers) following previous baseline recommendations (Wang et al., 2022). For confidence functions class $\mathcal{G}$, we use a class of 2-layer neural nets and provide the last two layers' representations from $h$ as input. We train it using SGD. The hyperparameters are deferred to Appendix C. Unless otherwise specified, our method uses pseudolabeling error tolerance $\epsilon = 5\%$.

### 5.2. Results and Discussion

To verify our main claims, we compare the baselines, their combinations with our method, and methods that induce calibrated scores in SSL. We run all methods with three random seeds and report (in Table 2) the mean and standard deviation of accuracy across three runs.

---

[1] https://github.com/harit7/PabLO-SSL

[2] We assign this name for convenience.

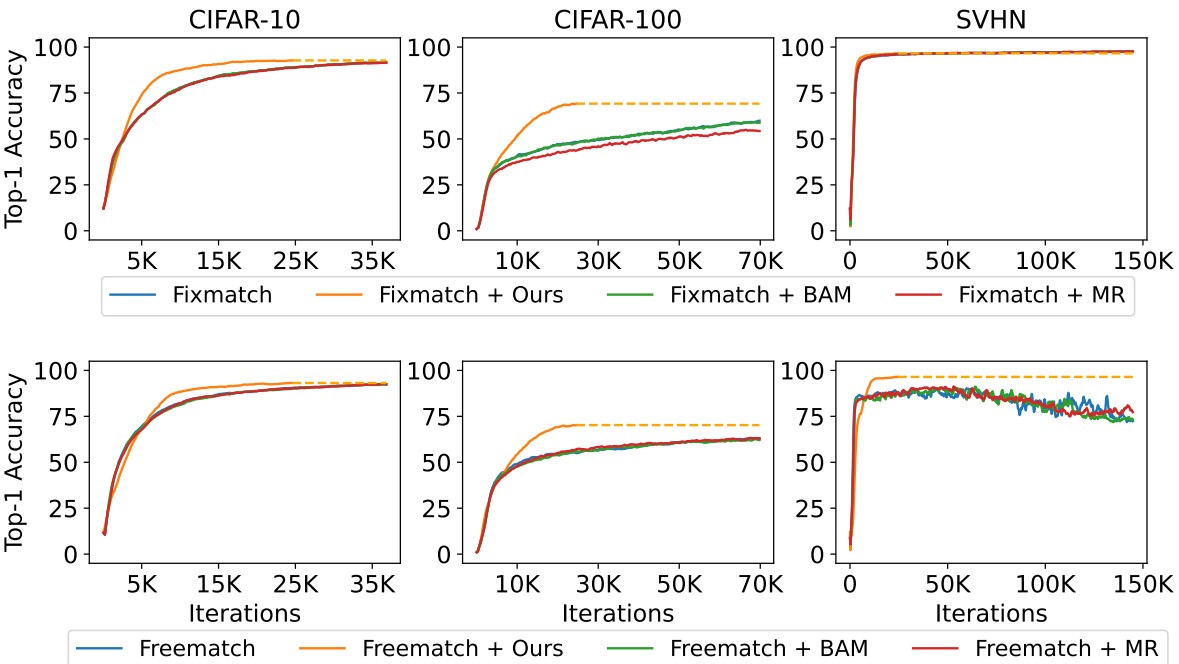

Figure 2. Top-1 accuracy of our method and baselines on CIFAR-10, CIFAR-100, and SVHN. We plot the values for every 200 steps.

**C1. Test accuracy improvements.** Since our method maximizes the pseudolabeling coverage and accuracy, it provides more accurate pseudolabels for model training. Therefore, we expect it to yield a model with better test accuracy than the baselines. We report the test accuracies at the end of 25K iterations in Table 2 for our methods. For the baselines, we report the test accuracies at the end of the corresponding adjusted number of iterations (well above 25K). Figure 2 illustrates how the top-1 accuracy evolves during the SSL. Similarly, Figure 4 and 5 show how batch pseudolabeling accuracy and batch pseudolabeling coverage change.

First, as expected, integrating our method into the base methods improves test accuracy across all settings. For CIFAR-10, using it with Fixmatch provides almost 2% improvement over Fixmatch alone, and using it with Freematch yields 1% improvement over Freematch. Much more significant improvements are observed in the much harder setting of CIFAR-100: a nearly 10% improvement in top-1 accuracy over Fixmatch and around 5% improvement over Freematch. SVHN is an easier setting; here, the improvements are marginal. With Fixmatch, our performance is similar to that of the baselines. But, using `PabLO` with Freematch improves the performance by 3%.

**C2. Error tolerance affects performance.** In our method, the error tolerance parameter $\epsilon$ is a knob to control the amount of noise in pseudolabels. A common wisdom in pseudolabeling is that higher noise will lead to worse performance, which is our expectation. To see this, we run our method with $\epsilon \in \{0.01, 0.05, 0.1, 0.2, 0.4\}$ in CIFAR-10

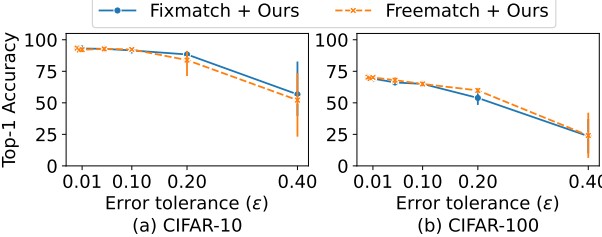

Figure 3. Top-1 accuracy of our method with different error tolerance $\epsilon$ on (a) CIFAR-10 and (b) CIFAR-100 dataset.

and CIFAR-100 settings, each with three random seeds, and report the results in Figure 3. The results are as expected — higher values of $\epsilon$ lead to degraded test accuracy due to high noise in the pseudolabels and with decreasing $\epsilon$ leads to improved accuracy. These results also suggest that prioritizing the quality (accuracy) of pseudolabels over quantity is a better choice in pseudolabeling. The results are also summarized in Table 8 and Table 9 in the Appendix.

To investigate the error tolerance further, we designed error tolerance scheduling using different error tolerances during various stages of SSL training. Table 3 summarizes the error tolerance we set at different iterations of SSL training and the corresponding top-1 accuracy for the CIFAR-10 and CIFAR-100 datasets. As we see, starting SSL with a small error tolerance and ending with a large tolerance severely impacts the performance on both CIFAR-10 and CIFAR-100 datasets. While our findings suggest a lower error tolerance is preferable, this may not hold in general. Nevertheless, our

| Dataset | CIFAR-10 | | CIFAR-100 | | SVHN | |
|---|---|---|---|---|---|---|
| # Labels | 40 | 250 | 400 | 2500 | 40 | 250 |
| Fixmatch | $57.16 \pm 12.12$ | $90.8 \pm 0.78$ | $30.38 \pm 0.68$ | $59.09 \pm 1.10$ | $62.27 \pm 5.57$ | $\mathbf{97.57 \pm 0.08}$ |
| Fixmatch + MR | $54.76 \pm 5.92$ | $90.41 \pm 0.83$ | $16.31 \pm 1.44$ | $54.16 \pm 0.18$ | $74.41 \pm 16.61$ | $97.55 \pm 0.08$ |
| Fixmatch + BaM | $57.72 \pm 1.44$ | $90.67 \pm 0.90$ | $31.45 \pm 1.29$ | $56.60 \pm 2.45$ | $76.18 \pm 18.61$ | $97.51 \pm 0.13$ |
| **Fixmatch + Ours** | $\mathbf{89.72 \pm 5.31}$ | $\mathbf{92.69 \pm 0.74}$ | $\mathbf{67.44 \pm 1.14}$ | $\mathbf{69.10 \pm 0.45}$ | $\mathbf{96.58 \pm 0.14}$ | $96.54 \pm 0.13$ |
| Freematch | $79.61 \pm 7.24$ | $92.26 \pm 0.18$ | $43.88 \pm 3.60$ | $63.13 \pm 0.46$ | $86.87 \pm 4.15$ | $92.90 \pm 2.76$ |
| Freematch + MR | $84.19 \pm 6.19$ | $92.17 \pm 0.36$ | $44.20 \pm 2.93$ | $62.03 \pm 0.82$ | $84.89 \pm 7.82$ | $93.26 \pm 2.36$ |
| Freematch + BaM | $88.33 \pm 0.33$ | $92.32 \pm 0.25$ | $44.49 \pm 3.05$ | $62.13 \pm 2.93$ | $81.43 \pm 14.80$ | $91.08 \pm 3.72$ |
| **Freematch + Ours** | $\mathbf{90.19 \pm 5.37}$ | $\mathbf{93.10 \pm 0.28}$ | $65.37 \pm 4.05$ | $68.76 \pm 1.38$ | $96.55 \pm 0.15$ | $\mathbf{96.65 \pm 0.26}$ |
| Softmatch | $83.60 \pm 7.09$ | $91.74 \pm 0.78$ | $40.73 \pm 1.46$ | $61.43 \pm 0.34$ | $59.52 \pm 12.47$ | $96.93 \pm 0.23$ |
| **Softmatch + Ours** | $\mathbf{89.96 \pm 4.74}$ | $\mathbf{93.14 \pm 0.33}$ | $\mathbf{67.84 \pm 0.33}$ | $68.74 \pm 0.72$ | $96.55 \pm 0.10$ | $96.51 \pm 0.37$ |
| Adamatch | $75.00 \pm 1.10$ | $91.35 \pm 0.66$ | $31.61 \pm 1.92$ | $58.08 \pm 0.44$ | $80.83 \pm 7.09$ | $96.99 \pm 0.08$ |
| **Adamatch + Ours** | $\mathbf{86.62 \pm 10.54}$ | $\mathbf{93.06 \pm 0.19}$ | $\mathbf{67.00 \pm 1.02}$ | $\mathbf{68.12 \pm 0.48}$ | $96.55 \pm 0.09$ | $96.44 \pm 0.20$ |

*Table 2.* Top-1 Accuracy for CIFAR-10, CIFAR-100 and SVHN averaged across 3 random seeds. We report the average accuracy $\pm$ std. deviation. Here, $N_l = \{40, 250\}$ for CIFAR-10, SVHN and $N_l = \{400, 2500\}$ for CIFAR-100. The best accuracy is **bolded.**

| | CIFAR-10 Top-1 Accuracy | CIFAR-100 Top-1 Accuracy |
|---|---|---|
| Schedule 1 Fixmatch + Ours | $91.11 \pm 1.31$ | $65.78 \pm 1.36$ |
| Schedule 1 Freematch + Ours | $91.09 \pm 1.01$ | $66.13 \pm 0.48$ |
| Schedule 2 Fixmatch + Ours | $35.38 \pm 29.27$ | $19.56 \pm 3.07$ |
| Schedule 2 Freematch + Ours | $30.48 \pm 10.52$ | $24.80 \pm 3.49$ |

| Method | Acc—True | Acc—False |
|---|---|---|
| Fixmatch | $67.62 \pm 2.10$ | $90.08 \pm 0.78$ |
| Fixmatch + MR | $64.78 \pm 4.64$ | $90.41 \pm 0.83$ |
| Fixmatch + BaM | $68.10 \pm 2.02$ | $90.67 \pm 0.90$ |
| Freematch | $85.40 \pm 1.36$ | $92.26 \pm 0.18$ |
| Freematch + MR | $83.59 \pm 2.59$ | $92.17 \pm 0.36$ |
| Freematch + BaM | $85.48 \pm 3.02$ | $92.32 \pm 0.25$ |
| **Fixmatch + Ours** | $\mathbf{92.69 \pm 0.74}$ | $\mathbf{92.80 \pm 0.56}$ |
| **Freematch + Ours** | $\mathbf{93.10 \pm 0.28}$ | $\mathbf{91.80 \pm 1.08}$ |

*Table 3.* Top-1 accuracy for the two error tolerance ($\epsilon$) scheduling. The table reports the $\epsilon$ we use between each iteration interval and the top-1 accuracy yielded by the corresponding schedule. For schedule 1, we set $\epsilon = 0.4, 0.2, 0.1, 0.05, 0.01$ when the number of iterations is in the following interval, respectively: $[0, 5K), [5K, 10K), [10K, 15K), [15K, 20K), [20K, 25K)$. For schedule 2, we set $\epsilon = 0.01, 0.05, 0.1, 0.2, 0.4$, for the same intervals, respectively.

*Table 4.* Results on CIFAR-10 with and without pseudolabel accumulation (Acc) for all the methods.

framework provides the flexibility to control this explicitly and thus can be tuned by practitioners for the setting at hand.

**C3. Is pseudolabel accumulation helpful?** Accumulation allows the methods to use old pseudolabels for points that couldn't get pseudolabeled in the current iteration. Thus, we expect accumulation to help improve the utilization of un-labeled data and lead to better test accuracy in cases where the pseudolabel quality is assured to be high in all iterations. We run two variations of our method and baselines — with and without accumulation and report the results in Table 4. We observe that our method has similar test accuracy irrespective of accumulation. However, accumulation achieves better coverage in early iterations, as observed

in Figure 6 in Appendix C. These results are unsurprising since our method ensures high quality of pseudolabels while maximizing coverage; it can eventually catch up with the version using accumulation, leading to similar final test accuracies. On the other hand, having accumulation hurts the performance of baseline models. This might be because the pseudolabels generated by the baseline models are inaccurate, especially in the earlier iterations, thus degrading the overall performance. Overall, we believe accumulation will be helpful when we have pseudolabels with high accuracy. The plots for pseudolabeling coverage and accuracy over the entire run are in Figures 6, 7 in Appendix C.

**Experiment with baselines using calibration data for training.** We run the baselines where the amount of training data for the baseline is increased by the amount of calibration data used in our method. For the CIFAR-10 setting with 250 labels, we run the baseline now with 250 + 1000 = 1250 labeled points for training. Similarly, for CIFAR-100 with 2500 labels setting, we run with 2500 + 3000 = 5500

| Method | CIFAR-10 | CIFAR-100 |
|---|---|---|
| FixMatch (train + cal) | $92.68 \pm 0.31$ | $64.77 \pm 0.10$ |
| FixMatch + Ours | $92.69 \pm 0.74$ | $\mathbf{69.10 \pm 0.45}$ |
| FreeMatch (train + cal) | $93.03 \pm 0.03$ | $67.69 \pm 0.12$ |
| FreeMatch + Ours | $93.10 \pm 0.28$ | $\mathbf{68.76 \pm 1.38}$ |

*Table 5.* Comparison of FixMatch and FreeMatch with and without our method on CIFAR-10 and CIFAR-100. Here $N_l = 250$ for CIFAR-10 and 2500 for CIFAR-100. The overall training data from methods annotated as (train + cal) includes these $N_l$ points and the calibration data $N_{\mathrm{cal}}$ (from Table 1) used in our method.

labels for training. The results are reported in the Table 5. We can see that even with more labeled data in training, the baselines still fall short significantly in the CIFAR-100 setting, while the performance gap in the CIFAR-10 (easier) setting narrows down.

**Validation data usage, limitations, and future work.** In the experiments, our method used parts of the validation data differently in comparison to the baselines. We comment on the role of validation data in the baselines, in our method, and potential limitations due to it. Recall, our goal is to address the problem of ad hoc choices of confidence scores and thresholds in pseudolabeling-based SSL. To this end, we introduced a principled solution to learn scores and thresholds that can directly achieve any specified pseudolabeling error while maximizing the number of pseudolabeled points. Given the focus of our paper, our experiments are designed to study whether using our learnable scores and thresholds can benefit the baselines. To keep our solution statistically sound, we used part of the validation data to learn the scores and thresholds. The lack (or cost) of a sufficient amount of validation data could be a limitation in practice. Future work can explore ways to reduce the amount of validation data needed for learning confidence scores and thresholds while preserving the soundness of our method. For instance, generative models or data augmentation techniques could be employed to this end.

More broadly, in semi-supervised learning (SSL) research, the cost of validation data is often not explicitly accounted for, partly due to the widespread use of standard benchmark datasets where hyperparameters have been tuned extensively over time. However, in real-world applications, new datasets typically require a substantial amount of validation data for model selection and hyperparameter tuning—both of which are essential parts of the modern training process. Developing novel benchmarks that explicitly incorporate validation data into the overall labeled data budget could help better reflect practical deployment settings and would be a valuable direction for future SSL research.

## 6. Conclusion

Common semi-supervised learning (SSL) methods rely on pseudolabeling, but their effectiveness is limited by unreliable confidence scores and heuristic thresholding strategies. We address these issues by introducing a principled framework for learning confidence scores and thresholds with explicit control over pseudolabeling error. We adapt existing SSL methods with this framework and empirically show that the adapted methods achieve a higher test accuracy compared to their standard versions. Additionally, we introduce pseudolabel accumulation and analyze its impact, showing that it benefits methods with reliable pseudolabels, such as those using our framework. In sum, by providing a principled, data-driven approach to obtaining scores and thresholds for pseudolabeling, our work enhances SSL methods and opens the door to more reliable and efficient pseudolabeling-based SSL.

## Impact Statement

This research improves semi-supervised learning, enabling more accurate and efficient machine learning in settings where labeled data is hard to obtain by following first principles in designing thresholds and confidence functions. Our work has various potential societal implications, with no specific concerns that require special attention in this context.

## Acknowledgments

This work was partly supported by funding from the American Family Data Science Institute and the Institute for Foundations of Data Science (IFDS). We thank the anonymous reviewers for their valuable feedback.

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

## Supplementary Material

The supplementary material is organized as follows. First, we summarize the notations in Table 6 in Appendix A, then we provide formal algorithms in Appendix B and additional experimental results and details are provided in Appendix C.

## A. Glossary

The notations are summarized in Table 6 below.

## B. Detailed Algorithms

---

**Algorithm 1** Estimate Pseudolabeling Thresholds Classwise

---

**Input:** Confidence function $\widehat{g}_i$, classifier $\widehat{h}_i$, Part of validation data $D_{\text{th}}^{(i)}$ for threshold estimation, pseudolabeling error tolerance $\epsilon$, space of thresholds $T$, label space $\mathcal{Y}$.
**Output:** Pseudolabeling thresholds $\widehat{\mathbf{t}}_i$, where $\widehat{\mathbf{t}}_i[y]$ is the threshold for class $y$.
**for** $y \in \mathcal{Y}$ **do**
    # Extract the set of points $D_{\text{th}}^{(y)}$ for which the groundtruth class is $y$.
    $D_{\text{th}}^{(y)} \leftarrow \{(\mathbf{x}', y') \in D_{\text{th}} : y' = y\}$
    $T_y' \quad\leftarrow T \cup \{\infty\}$.
    # Estimate pseudolabeling error at each threshold on class specific data $D_{\text{th}}^{(y)}$. Pick the smallest threshold with the sum
    of the estimated error and $C_1$ times the std. deviation is below $\epsilon$. Here $C_1$ is set to 0.25 and $\widehat{\sigma}(z) = \sqrt{z(1-z)}$.
    $\widehat{\mathbf{t}}_i[y] \quad\leftarrow \min\{t \in T_y' : \widehat{\mathcal{E}}(\widehat{g}_i, t \mid \widehat{h}_i, D_{\text{th}}^{(y)}) + C_1\widehat{\sigma}(\widehat{\mathcal{E}}(\widehat{g}_i, t \mid \widehat{h}_i, D_{\text{th}}^{(y)})) \leq \epsilon\}$,
**end for**
**return** $\widehat{\mathbf{t}}_i$

---

---

**Algorithm 2** Estimate Pseudolabeling Threshold Jointly for All Classes

---

**Input:** Confidence function $\widehat{g}_i$, classifier $\widehat{h}_i$, Part of validation data $D_{\text{th}}^{(i)}$ for threshold estimation, pseudolabeling error tolerance $\epsilon$, space of thresholds $T$, label space $\mathcal{Y}$.
**Output:** Pseudolabeling thresholds $\widehat{\mathbf{t}}_i$, where $\widehat{\mathbf{t}}_i[y]$ is the threshold for class $y$.
$T' \leftarrow T \cup \{\infty\}$
# Estimate pseudolabeling error at each threshold on the entire set $D_{\text{th}}$. Pick the smallest threshold with the sum of the
estimated error and $C_1$ times $\widehat{\sigma}$ is below $\epsilon$. Here $C_1$ is set to 0.25 and $\widehat{\sigma}(z) = \sqrt{z(1-z)}$.
$t \leftarrow \min\{t \in T' : \widehat{\mathcal{E}}(\widehat{g}_i, t \mid \widehat{h}_i, D_{\text{th}}) + C_1\widehat{\sigma}(\widehat{\mathcal{E}}(\widehat{g}_i, t \mid \widehat{h}_i, D_{\text{th}})) \leq \epsilon\}$.
**for** $y \in \mathcal{Y}$ **do**
    $\widehat{\mathbf{t}}_i[y] \leftarrow t$
**end for**
**return** $\widehat{\mathbf{t}}_i$

---

---

**Algorithm 3** Pseudolabeling Based SSL with `PabLO`

---

**Input:** Labeled data for training $D_l$, validation data $D_{\mathrm{val}}$, unlabeled pool $X_u$, error tolerance $\epsilon$, use-accumulation flag, num_iters, batch size $B$, replication factor $\mu$, weak $\omega$ and strong $\Omega$ augmentations, number of calibration points $N_{\mathrm{cal}}$, num. of threshold estimation points $N_{\mathrm{th}}$, frequency of invoking `PabLO` $F$, space of thresholds $T$, label space $\mathcal{Y}$.

**Output:** $\widehat{h}_{\mathrm{ssl}}$, model with the best validation accuracy.

  # Set initial pseudolabels and masks to 0.

  $\widetilde{Y}_u^{(0)} \leftarrow [0, 0, \ldots, 0], S_u^{(0)} \leftarrow [0, 0, \ldots, 0], i \leftarrow 1$

  # Draw calibration and threshold estimation sets from $D_{\mathrm{val}}$.

  $D_{\mathrm{cal}}, D_{\mathrm{th}} \leftarrow \texttt{DrawRandomly}(D_{\mathrm{val}}, N_{\mathrm{cal}}, N_{\mathrm{th}})$.

  # Training loop with pseudolabeling.

  **while** $i \leq$ num_iters **do**

    # Draw batches $D_l^b$, $X_u^b$ of labeled and unlabeled points, $I_u^b$ denotes the indices corresponding to points in $X_u^b$.

    $D_l^b, X_u^b, I_u^b \leftarrow \texttt{DrawRandomBatch}(\mu D_l, \mu X_u, B)$

    # Create weak and strong augmentations of $X_u^b$.

    $X_{u,w}^b, X_{u,s}^b \leftarrow \omega(X_u^b), \Omega(X_u^b)$

    /** Begin Pseudolabeling Block **/

    # Perform pseudolabeling using `PabLO`.

    **if** $i\%F = 0$ **then**

      # Get $\widehat{g}_i$ by solving optimization (P1).

      $\widehat{g}_i, \widehat{\mathbf{t}}_i' \leftarrow \texttt{SolveOptProblemP1}(\widehat{h}_i, D_{\mathrm{cal}})$

      # Estimate pseudolabeling thresholds.

      **if** estimate threshold classwise **then**

        # Use Algorithm 1.

        $\widehat{\mathbf{t}}_i \leftarrow \texttt{ClasswiseThreshold}(\widehat{g}_i, \widehat{h}_i, D_{\mathrm{th}}, \epsilon, T, \mathcal{Y})$

      **else**

        # Use Algorithm 2.

        $\widehat{\mathbf{t}}_i \leftarrow \texttt{JointThreshold}(\widehat{g}_i, \widehat{h}_i, D_{\mathrm{th}}, \epsilon, T, \mathcal{Y})$

      **end if**

      # Compute fresh psuedolabels $\widetilde{Y}_u^{(i)}$ and pseudolabeling masks $S_u^{(i)}$ for all points in $X_u$.

      $\widetilde{Y}_u^{(i)} \leftarrow \widehat{h}_i(\omega(X_u)), \quad S_u^{(i)} \leftarrow \mathbb{1}(\widehat{g}_i(\omega(X_u)) \geq \widehat{\mathbf{t}})$

      **if** use-accumulation **then**

        # Apply pseudolabel accumulation if enabled.

        $\widetilde{Y}_u^{(i)} \leftarrow S_u^{(i)} \widetilde{Y}_u^{(i)} + (1 - S_u^{(i)}) \widetilde{Y}_u^{(i-1)}$

        $S_u^{(i)} \leftarrow S_u^{(i)} \vee S_u^{(i-1)}$

      **end if**

    **else**

      $\widehat{g}_i, \widehat{\mathbf{t}}_i = \widehat{g}_{i-1}, \widehat{\mathbf{t}}_{i-1}$

    **end if**

    /** End Pseudolabeling Block **/

    # Extract pseudolabels and masks for the current unlabeled batch. Then compute supervised and unsupervised losses.

    $\widetilde{Y}_u^b, S_u^b \leftarrow \widetilde{Y}_u[I_u^b], \ S_u[I_u^b]$

    $\widehat{\mathcal{L}}_s(\widehat{h}_i) \leftarrow \texttt{supervised\_loss}(h, D_l^b)$

    $\widehat{\mathcal{L}}_u(\widehat{h}_i) \leftarrow \texttt{unsupervised\_loss}(h, X_{u,w}^b X_{u,s}^b, \widetilde{Y}_u^b, S_u^b)$

    $\widehat{\mathcal{L}}(\widehat{h}_i) \leftarrow \widehat{\mathcal{L}}_s(\widehat{h}_i) + \lambda_u \widehat{\mathcal{L}}_u(\widehat{h}_i)$

    # Perform a gradient descent step to get new model $\widehat{h}_{i+1}$.

    $\widehat{h}_{i+1} \leftarrow \texttt{SGD\_update}(\widehat{\mathcal{L}}(\widehat{h}_i)); \quad i \leftarrow i + 1$

    # Evaluate model on $D_{\mathrm{val}}$ to keep track of the best model.

    **if** $i\%$eval_freq $= 0$ **then**

      eval_acc $\leftarrow \texttt{evaluate\_model}(\widehat{h}_i, D_{\mathrm{val}})$

      If eval_acc is best so far then $\widehat{h}_{\mathrm{ssl}} = \widehat{h}_i$

    **end if**

  **end while**

---

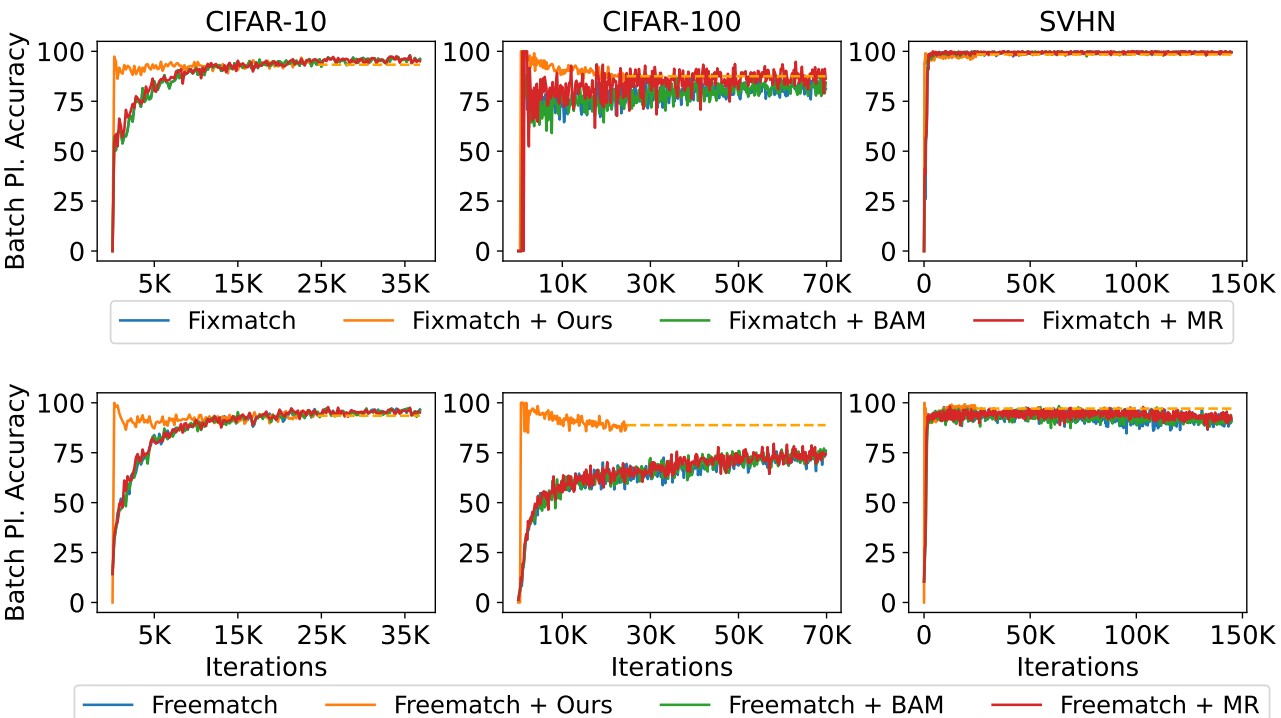

*Figure 4.* Batch pseudolabel accuracy of our method and baselines on CIFAR-10, CIFAR-100, and SVHN. We plot the values for every 200 steps.

## C. Additional Experiments and Details

**Compute.** We ran all of our experiments on a high-throughput system with various GPUs. Therefore, each individual experiment task may be scheduled among NVIDIA A100 SXM4-40GB, NVIDIA A100 SXM4-80GB, NVIDIA L40, and NVIDIA H100 80GB HBM3. We measured the runtime of our algorithm on a desktop with a single NVIDIA RTX 4090. On CIFAR-10, it took about 0.203 seconds for each iteration for our method and around 0.140 seconds for the baselines. On CIFAR-100, it took about 0.396 seconds for each iteration for our method and around 0.143 seconds for the baselines. On SVHN, it took about 1.275 seconds for each iteration for our method and around 0.225 seconds for the baselines.

**Hyperparameters.** For the baselines, we have used their default settings. To maintain consistency and experiment the efficiency of method, we used WRN-28-2 which is 1.4M parameter model for all the datasets. We summarize the main hyperparameters we have used in our method in Table 10.

Note that the number of epochs we used to train the function $g$ and to estimate $\mathbf{t}$ is dynamic. That is, its actual value depends on and is proportional to the current number of iterations of the SSL training. More concretely, at iteration $i$ of SSL training, we use $\min(\lfloor i/25 \rfloor, \text{max epoch})$ number of epochs to find $g$ and $\mathbf{t}$.

We additionally conduct the following ablation study to study our technique's dependence on the amount of data used in learning $g$ and thresholds.

**A2. How much data is needed to learn the $g$ and $\mathbf{t}$?** We take $N_{\text{cal}}$ and $N_{\text{th}}$ from the validation data to learn the confidence function $g$ and estimate the thresholds $\mathbf{t}$ respectively. Intuitively larger values of these should lead to good $g$ and $\mathbf{t}$ that can extract the expected level of pseudolabeling coverage and accuracy from the classifier at hand. However, the task of learning good $g$ and estimating thresholds is not super hard and we expect it will take fewer samples to be successful. To understand this better we run our method with $N_{\text{cal}}$ and $N_{\text{th}}$ in $\{250, 500, 750, 1000\}$ on CIFAR-10 setting for 3 random seeds and report the result in Fig 8. We observe that our method can achieve desired performance with just 500 labeled points (i.e 50 labels per class). This is interesting because we can achieve 90% accuracy by just using 250 points ($N_l$) for training $h$ and a total of 1K for learning $g$. Refer to Table 7 for more details.

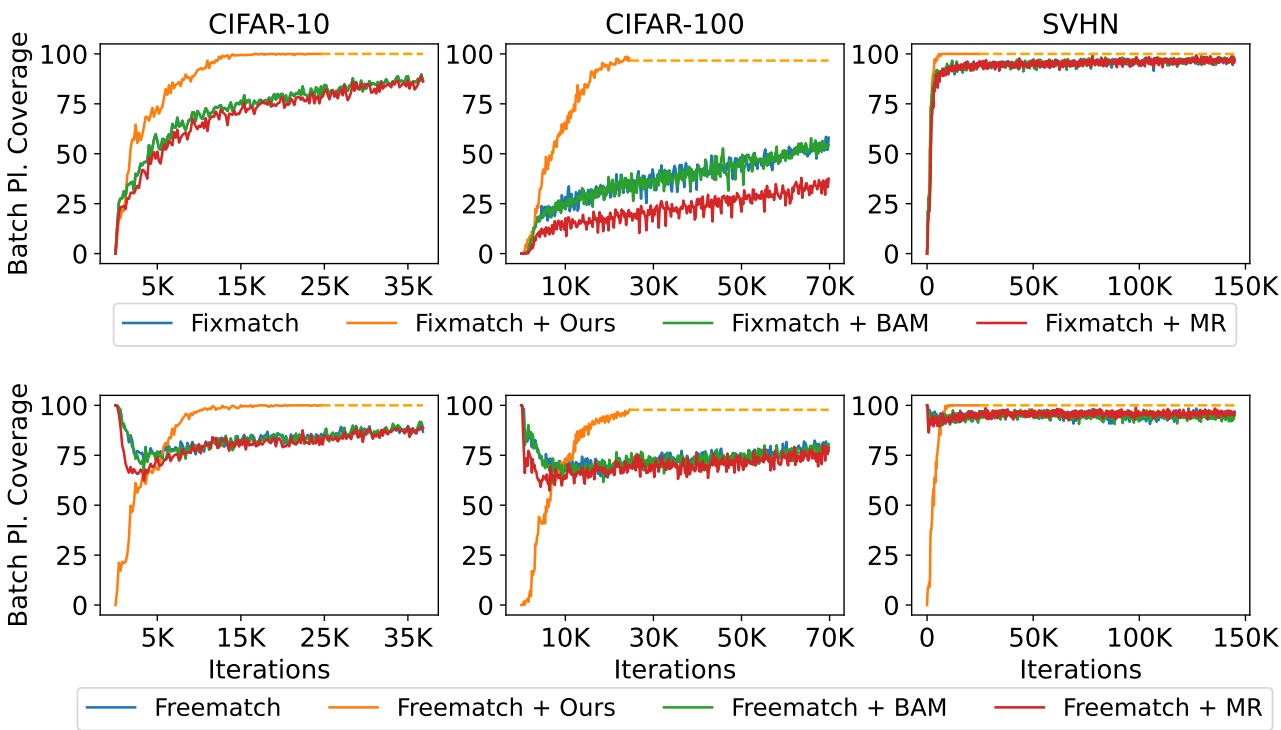

*Figure 5.* Batch pseudolabel coverage of our method and baselines on CIFAR-10, CIFAR-100, and SVHN. We plot the values for every 200 steps.

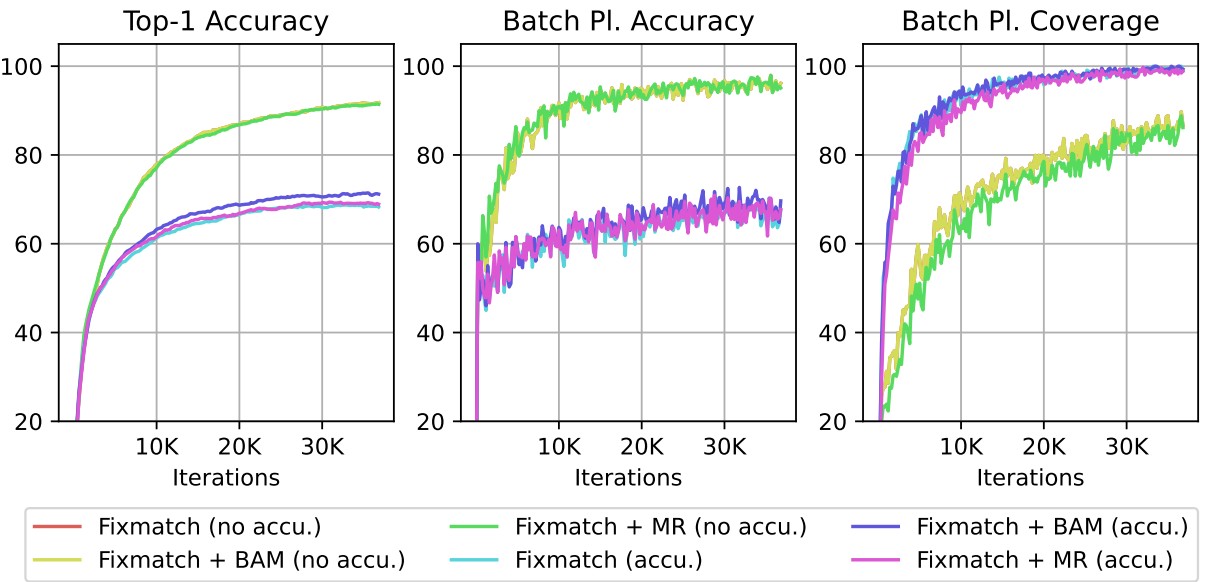

*Figure 6.* **(A1.)** Left to Right: Top-1 Accuracy, Batched pseudolabeling Accuracy, and batched pseudolabeling coverage of Fixmatch with and without pseudolabeling accumulation enabled on CIFAR-10. It can be seen that enabling pseudolabeling accumulation worsen the performance of baseline methods in terms of accuracy and coverage.

| Symbol | Definition |
|---|---|
| $\mathcal{X}$ | feature space. |
| $\mathcal{Y}$ | label space i.e. $1, 2, \ldots k$. |
| $\mathcal{H}$ | hypothesis space (model class for the classifiers). |
| $\mathcal{G}$ | space of confidence functions. |
| $k$ | number of classes. |
| $\mathbf{x}, y$ | $\mathbf{x}$ is a datapoint in $\mathcal{X}$ and $y$ is its true label (if available). |
| $h$ | a model $h : \mathcal{X} \to \mathcal{Y}$. |
| $g$ | confidence function $g : \mathcal{X} \to T^k \subseteq \mathbb{R}^k$ |
| $\widehat{y}$ | hard label prediction. |
| $\widetilde{y}$ | $\widehat{y}$ is used as pseudolabel. |
| $\widehat{h}_{\text{ssl}}$ | a best learned model using SSL. |
| $\epsilon$ | pseudolabeling error tolerance. |
| $g_i^\star$ | optimal confidence function at $i$ iteration. |
| $\mathbf{t}_i^*$ | optimal threshold at $i$ iteration. |
| $X_u$ | available unlabeled data drawn from the distribution $P_x$ over $\mathcal{X}$. |
| $X_u^b$ | batch of unlabeled data. |
| $D_l$ | set of labeled data points drawn from the distribution $P_{xy}$. |
| $D_l^b$ | batch of labeled data. |
| $D_{\text{val}}$ | validation data. |
| $D_{\text{cal}}$ | calibration data; part of validation data used to optimize surrogate functions. |
| $D_{\text{th}}$ | part of validation data to estimate threshold $\mathbf{t}$. |
| $\mathbf{t}$ | $k$ dimensional vector of thresholds representing for $k$ classes. |
| $\mathbf{t}[y]$ | $y$th entry of $\mathbf{t}$ i.e. the threshold for class $y$. |
| $n_u$ | number of unlabeled points, i.e. size of $X_u$ used for consistency regularization and pseudolabeling. |
| $N_l$ | number of labeled points, i.e. size of $D_l$. Usual SSL setting has, $N_l \ll n_u$. |
| $N_{\text{val}}$ | number of points used for model selection. |
| $N_{\text{test}}$ | number of test data points. |
| $N_{\text{cal}}$ | number of points used for learning the $g$ function. |
| $N_{\text{th}}$ | number of data points used for threshold estimation. |
| $\widehat{\mathcal{L}}_s$ | supervised loss. |
| $\widehat{\mathcal{L}}_u$ | unsupervised loss with weighted importance $\lambda_u$. |
| $\widehat{\mathcal{L}}_r$ | sum of regularization terms for supervised and unsupervised loss with weighted importance $\lambda_r$. |
| $H(y, h, \mathbf{x})$ | standard cross-entropy loss. |
| $S(\mathbf{x}, g, \mathbf{t} \mid h)$ | pseudolabeleing mask. |
| $\omega$ | weak transformation, $\omega : \mathcal{X} \mapsto \mathcal{X}$. |
| $\Omega$ | strong transformation, $\Omega : \mathcal{X} \mapsto \mathcal{X}$. |
| $\alpha_o, \alpha_b$ | average time taken by our method and baseline methods. |
| $\widehat{\mathcal{P}}(g, \mathbf{t} \mid h, X)$ | estimated pseudolabeling coverage, see eq. (1). |
| $\mathcal{P}(g, \mathbf{t} \mid h)$ | population level pseudolabeling coverage, see eq. (2). |
| $\widehat{\mathcal{E}}(g, \mathbf{t} \mid h, D)$ | estimated pseudolabeling error, see eq. (3). |
| $\mathcal{E}(g, \mathbf{t} \mid h)$ | population level pseudolabeling error, see eq. (4). |
| $\widetilde{\mathcal{P}}(g, \mathbf{t} \mid h, D)$ | surrogate estimated pseudolabeling coverage, see eq. (6). |
| $\widetilde{\mathcal{E}}(g, \mathbf{t} \mid h, D)$ | surrogate estimated pseudolabeling error, see eq. (7). |
| $\lambda$ | hyperparamter controlling the importance of pseudolabeleing coverage and error in (P1). |

*Table 6.* Glossary of variables and symbols used in this paper.

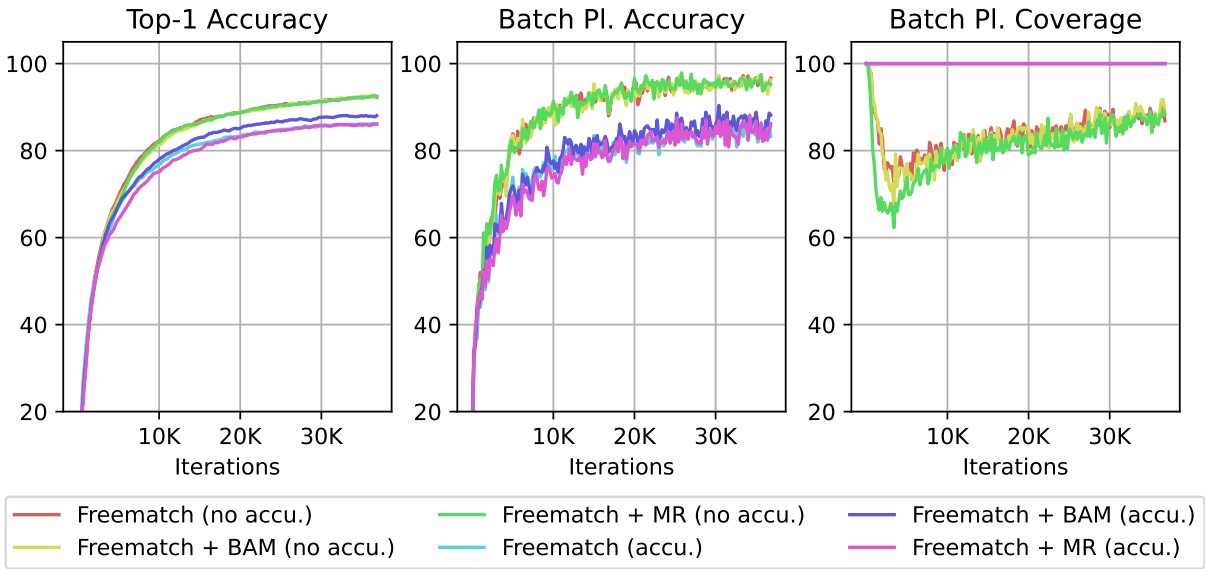

*Figure 7.* **(A1.)** Left to Right: Top-1 Accuracy, Batched pseudolabeling Accuracy, and batched pseudolabeling coverage of Freematch with and without pseudolabeling accumulation enabled on CIFAR-10. It can be seen that enabling pseudolabeling accumulation worsen the performance of baseline methods in terms of accuracy and coverage.

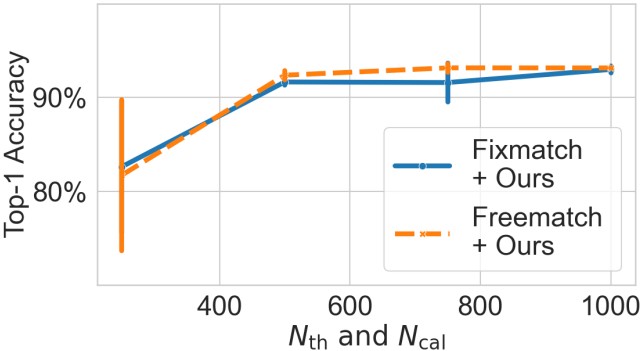

*Figure 8.* Top-1 accuracy of our method with different $N_{\text{th}}$ and $N_{\text{cal}}$.

| Method | $N_{\text{cal}} = N_{\text{th}} = 250$ | $N_{\text{cal}} = N_{\text{th}} = 500$ | $N_{\text{cal}} = N_{\text{th}} = 750$ |
|---|---|---|---|
| Fixmatch + Ours | $82.67 \pm 7.08$ | $91.74 \pm 0.41$ | $91.66 \pm 2.11$ |
| Freematch + Ours | $82.13 \pm 7.93$ | $92.33 \pm 0.49$ | $93.20 \pm 0.53$ |

*Table 7.* Results on CIFAR-10 with varying $N_{\text{cal}}$ and $N_{\text{th}}$.

| Method | $\epsilon = 0.01$ | $\epsilon = 0.1$ | $\epsilon = 0.2$ | $\epsilon = 0.4$ |
|---|---|---|---|---|
| Fixmatch + Ours | $93.05 \pm 0.54$ | $91.54 \pm 0.95$ | $88.35 \pm 2.90$ | $56.72 \pm 22.25$ |
| Freematch + Ours | $92.11 \pm 1.18$ | $92.31 \pm 0.16$ | $83.89 \pm 10.36$ | $52.17 \pm 25.36$ |

*Table 8.* Results on CIFAR-10 with varying $\epsilon$.

| Method | $\epsilon = 0.01$ | $\epsilon = 0.1$ | $\epsilon = 0.2$ | $\epsilon = 0.4$ |
|---|---|---|---|---|
| Fixmatch + Ours | $69.19 \pm 1.13$ | $65.01 \pm 0.34$ | $53.88 \pm 8.15$ | $23.58 \pm 18.21$ |
| Freematch + Ours | $70.13 \pm 0.67$ | $64.95 \pm 1.41$ | $59.83 \pm 1.32$ | $24.09 \pm 17.22$ |

*Table 9.* Results on CIFAR-100 with varying $\epsilon$.

| Method | Hyperparameter | Values |
|---|---|---|
| Learning $g$ function | optimizer | SGD |
| | learning rate | 0.01 |
| | batch size | 64 |
| | max epoch | 500 |
| | weight decay | 0.01 |
| | momentum | 0.9 |
| Estimating $\mathbf{t}$ | optimizer | SGD |
| | learning rate | 0.01 |
| | batch size | 64 |
| | max epoch | 500 |
| | weight decay | 0.01 |
| | momentum | 0.9 |

*Table 10.* Hyperparameters used for our method.

