# OpenReview forum: "Rethinking Confidence Scores and Thresholds in Pseudolabeling-based SSL"
_ICML.cc/2025/Conference — ICML 2025 poster_

### Official Review · Reviewer_sFAy · 2025-02-17

**Overall Recommendation:** 3

**Summary:**

This paper proposes a method for selection of points to be pseudolabeled in pseudolabeling-based semi-supervised learning idea. Contrasting previous works which use confidence-based thresholding, PaBlo trains a selector function with an optimization objective which balances coverage with pseudolabeling error, using a subset of the held-out validation data.

Update after rebuttal:
Apologies for the delay. Thanks for the additional results and new experiments. I'm raising my score to a Weak accept (3).
However, I think the paper should incorporate an extensive discussion on the data availability assumptions that this method makes (validation data).

**Claims And Evidence:**

Claim 1: the adaptations of popular pseudolabelingbased SSL methods with PabLO output models with better test accuracy.

This claim is verified on 3 datasets (CIFAR 10, CIFAR 100 and SVHN) with baselines Freematch and FixMatch.
It would be interesting to include additional baselines which do not solely rely on simple confidence thresholds, such as uncertainty-based methods (e.g. [1], [2])


[1] Rizve, M. N., Duarte, K., Rawat, Y. S., & Shah, M. (2021). In defense of pseudo-labeling: An uncertainty-aware pseudo-label selection framework for semi-supervised learning. arXiv preprint arXiv:2101.06329.

[2] Nguyen, V., Husain, H., Farfade, S., & Hengel, A. V. D. (2022). Confident sinkhorn allocation for pseudo-labeling. arXiv preprint arXiv:2206.05880.

My main question about this experimental section is about the data splits. Given the statistics in Table 1, it seems that the validation set (which is labeled) used for the training of the selector function is much bigger than the labeled dataset used for training (6K samples, while the labeled data is only 250 samples for CIFAR 10), which contradicts the main purpose of semi-supervised learning, where labeled data is scarce. This brings a natural question: what happens if you use the validation data as labeled training data, and just use one of the baseline methods? I expect the performance of the baselines to increase significantly by using this validation data naively.  Furthermore, the results in Table 6, for N_cal = N_th = 250 (approximately 82% top-1 accuracy) seem much lower than the results of the baselines Fixmatch and Freematch reported in Table 2 for CIFAR 10 (90.8% and 92.26%).

Claim 2: a lower error tolerance is preferable

It would be interesting to report the number of points which are pseudolabeled in Figure 3, and smaller values of epsilon. Indeed, with 0 error tolerance, we should recover classical supervised learning, which should be worse than pseudo-labeling. Hence it would be more intuitive to see an "inverse" U shape in Figure 3.

Claim 3: Accumulation is not always helpful

This claim is supported with results on CIFAR-10.

**Essential References Not Discussed:**

The related work section covers the main strands of prior works related to this paper.

**Experimental Designs Or Analyses:**

See above:  My main question about this experimental section is about the data splits. Given the statistics in Table 1, it seems that the validation set (which is labeled) used for the training of the selector function is much bigger than the labeled dataset used for training (6K samples, while the labeled data is only 250 samples for CIFAR 10), which contradicts the main purpose of semi-supervised learning, where labeled data is scarce. This brings a natural question: what happens if you use the validation data as labeled training data, and just use one of the baseline methods? I expect the performance of the baselines to increase significantly by using this validation data naively.  Furthermore, the results in Table 6, for N_cal = N_th = 250 (approximately 82% top-1 accuracy) seem much lower than the results of the baselines Fixmatch and Freematch reported in Table 2 for CIFAR 10 (90.8% and 92.26%).

**Methods And Evaluation Criteria:**

The datasets are commonly used in the PL litterature.
Other baselines could be included to compare with Pablo.

**Other Comments Or Suggestions:**

NA

**Other Strengths And Weaknesses:**

The paper is well-written and easy to follow.

**Questions For Authors:**

See above on the data splits + the results in Table 6.

**Relation To Broader Scientific Literature:**

This paper positions itself as an improvement over the confidence-based PL methods which use thresholding.

**Theoretical Claims:**

NA

---

> ### Author Rebuttal · Authors · 2025-04-01
>
> We appreciate the feedback and questions. Our response is as follows,
>
> **On baselines that do not rely on confidence scores and thresholds.**
> While this would be interesting, our paper's focus is on pseudolabeling methods based on confidence scores and thresholds. For this reason, we chose baselines necessary to evaluate our claim — using our scores and thresholds in the common methods (FixMatch, FreeMatch) can improve the performance significantly, in contrast to using them with ad-hoc choices of scores and thresholding techniques. We also compared the variations of these with recent methods (BAM, MR) designed to encourage calibrated scores in SSL settings.
>
>
> **On validation data requirements.** We clarify as follows.
>
> * *it is common practice in SSL methods to use validation data to find the best model checkpoint by evaluating validation accuracy*. The baselines use all $N_{val}$ validation samples for this purpose.
>
> * In our method, we split this $N_{val}$ data into three parts, $N_{val}'$, $N_{cal}$, $N_{th}$ and use $N_{val}'$ for the usual model checkpoint evaluation and use $N_{cal}, N_{th}$ for learning confidence function and estimating thresholds respectively. **Note $N_{cal}$ and $N_{th}$ are much smaller than $N_{val}$**.
>
> * The idea of using the validation data for training is natural. Common benchmarks have been extensively studied in the literature for which there is a reasonable understanding of hyperparameters and expected performance. However, this overlooks the generality of SSL methods. In general, most of the SSL methods need validation data for model selection, tuning hyperparameters, etc. This cost is often overlooked in the prior works. Our work mentions it transparently.
>
> **Clarification on Table 6 results.** These results are to study the effect of sizes of calibration $N_{cal}$ and threshold estimation sets $N_{th}$. When these are too small, it will result in high variance in learning scores and thresholds, resulting in unreliable pseudolabeling (high excess pseudolabeling error). Using uniform convergence results it can be shown that the excess pseudolabeling error will scale as $O(\frac{1}{\sqrt{N_{cal}}} +\frac{1}{\sqrt{N_{th}}})$. Thus we see as $N_{cal}, N_{th}$ increases the performance gets better.
>
> **Inverse U Shape curve for error tolerance.**  In general, we expect an inverse U shape for test accuracy vs epsilon curve. In the Fig. 3. setting, we did not see this. We have run this experiment in one more setting where we see the inverse U shape curve. We summarize our findings in two points,
>
> * We note that zero error tolerance is not equivalent to classical supervised learning. We see sizeable pseudolabeling taking place even with 0 error tolerance in certain settings, e.g., as in Figure 3, CIFAR-10 with 250 labeled points for training. **[Please see the new results here](https://anonymous.4open.science/r/icml-rebuttal-2024-anon-E247/cifar10_low_high_epsilon.png)** . This means reasonable models are found early in the training that are 100% accurate on part of the space and our scores, and thresholds are able to find that space.
>
> * Second, in settings with an even smaller amount of labeled data for training, we start to see the expected inverse U shape curve. We ran the procedure in the CIFAR-10 setting with 40 labeled points for training with various choices of $\epsilon \in$  { 0.1%, 1%, 5%, 20%, 40% }. **[See new results](https://anonymous.4open.science/r/icml-rebuttal-2024-anon-E247/cifar10_epsilon.png)** for the plot of test accuracy corresponding to each $\epsilon$. The test accuracies are 74.6%, 91.5%, and 83.8% corresponding to $\epsilon$ 0.1%, 1%, and 5% respectively.
>
>
> We hope our response resolves your queries. We are happy to answer any further questions.

---

> > ### Comment · Reviewer_sFAy · 2025-04-04
> >
> > Thank you for your detailed rebuttals. I appreciate the clarification on Table 6 and the new results for the inverse U shape curve.
> > I still have questions regarding the other 2 points.
> >
> > **Regarding validation data requirements**: I acknowledge that, as you mention, validation data is widely used in the SSL litterature. While it is quite unrealistic in real-worlds settings to have such a big validation labeled dataset, it is understandable from a benchmarking perspective.
> > However, I am not entirely convinced in your case about this assumption of using directly such held-out  data in training the pipeline.
> > Indeed, based on Table 1, in CIFAR 10, the calibration dataset is 4 times bigger than the labeled dataset (1K vs 250). And such a calibration dataset (which is labeled) *directly impacts the learning process*, as it is used to train the surrogate models. This is quite different from the use of validation data for *benchmarking purposes*, where the validation data is only used at the end to select the model (or do HPT tuning). Hence a natural question is: what happens if you leverage directly this calibration data (1K samples) as labeled examples in FixMatch/FlexMatch. In practice, this would require the same number of labeled samples at training time than your proposed method, hence makes it as applicable as your method.
> >
> > **Regarding baselines that do not rely on confidence scores and thresholds**: I acknowledge the contribution of this paper as an improvement over confidence based pseudo-labeling. However, I still think it is important to get results for at least one competitive PL method that also uses uncertainty (which claims to be better than confidence-based PL). This will allow to see the performance gap, and if your method narrows this gap. Else, it is complicated to see the value of using a confidence-based PL method (even improved with your method) vs other baselines.

---

> > > ### Author Response · Authors · 2025-04-09
> > >
> > > Thanks for the comment, and we are glad that some of your concerns are resolved. We answer the remaining queries below.
> > >
> > > **Note on our goal and experimental setup.**  Our goal is to address the *problem of ad-hoc choices of confidence scores and thresholds in pseudolabeling-based SSL*. To this end, we proposed a *principled solution to learn scores and thresholds* that can directly achieve any specified pseudolabeling error while maximizing the number of pseudolabeled points.  Given the focus of our paper, our experiments are designed to study whether using our learnable scores and thresholds can benefit the baselines. To keep our solution statistically sound, we used part of the validation data to learn the scores and thresholds.
> > >
> > > **General comments on validation data.** We make two points on the role of validation data.
> > >
> > > a) The real cost of validation data is grossly underestimated (often overlooked) in the prior works since the focus has been on common benchmark datasets where the hyperparameters have been tuned extensively over a long period of time.
> > >
> > > b) In general, for a new application (dataset), one would require a non-trivial amount of validation data for model selection and hyperparameter tuning. Note, these are all part of the model training process. Introducing novel datasets and benchmarks accounting for the cost of validation data in the overall labeled data can be useful for the SSL research in general and would be a fruitful direction for future work.
> > >
> > >
> > > **Experiment with baselines using calibration data for training.**
> > > As suggested, we run the baselines where the amount of training data for the baseline is increased by the amount of calibration data used in our method. So, for CIFAR-10 setting in the paper with 250 labels, we run the baseline now with 250 + 1000 = 1250 labeled points for training. Similarly, for CIFAR-100 with 2500 labels setting, we run with 2500 + 3000 = 5500 labels for training. The results are reported in the table below. The baselines using calibration data for training are annotated with ( train+cal ). We can see that even with more labeled data in training, the baselines still fall short significantly in the CIFAR-100 setting, while the performance gap in the CIFAR-10 (easier) setting narrows down.
> > >
> > >
> > > | Dataset          | CIFAR-10    | CIFAR-100  |
> > > |------------------|--------------|--------------|
> > > | Fixmatch (train + cal)       | 92.68 ± 0.31 | 64.77 ± 0.10 |
> > > | Fixmatch + Ours  | 92.69 ± 0.74 | **69.10 ± 0.45** |
> > > | Freematch (train + cal)        | 93.03 ± 0.03 | 67.69 ± 0.12 |
> > > | Freematch + Ours | 93.10 ± 0.28 | **68.76 ± 1.38** |
> > >
> > >
> > >
> > > **Comparison with the suggested UPS [1] baseline.**
> > > We compare against the suggested baseline UPS [1] and additional baselines, Softmatch [3], and Adamatch [2]. The results (below tables) remain consistent with the claims in the main paper, i.e., using our scores and thresholds in the baselines improves their performance, and we see UPS falls short in comparison to other baselines. We will include these results in the paper.
> > >
> > >
> > > The following table corresponds to $N_l = 250$ for Cifar-10 and $2500$ for CIFAR-100.
> > >
> > > | Dataset          | CIFAR-10     | CIFAR-100    |
> > > |------------------|--------------|--------------|
> > > | Softmatch        | 91.74 ± 0.78 | 61.43 ± 0.34 |
> > > | Softmatch + Ours | **93.14 ± 0.33** | **68.74 ± 0.72** |
> > > | Adamatch         | 91.35 ± 0.66 | 58.08 ± 0.44 |
> > > | Adamatch + Ours  | **93.06 ± 0.19** | **68.12 ± 0.48** |
> > > | UPS              | 64.32        | 37.33        |
> > >
> > >
> > > The following table corresponds to $N_l = 40$ for Cifar-10 and $400$ for CIFAR-100.
> > >
> > > | Dataset          | CIFAR-10      | CIFAR-100    |
> > > |------------------|---------------|--------------|
> > > | Softmatch        | 83.60 ± 7.09  | 40.73 ± 1.46 |
> > > | Softmatch + Ours | **89.96 ± 4.74**  | **67.84 ± 0.33** |
> > > | Adamatch         | 75.00 ± 1.10  | 31.61 ± 1.92 |
> > > | Adamatch + Ours  | **86.62 ± 10.54** | **67.00 ± 1.02** |
> > > | UPS              | 20.30         | 9.34         |
> > >
> > > -----
> > >
> > > [1] Rizve et al.,  In Defense of Pseudo-labeling: An Uncertainty-Aware Pseudo-Label Selection Framework For Semi-Supervised Learning, ICLR, 2021.
> > >
> > > [2] Berthelot et al.,  AdaMatch: A Unified Approach to Semi-Supervised Learning and Domain Adaptation, ICLR, 2022.
> > >
> > > [3] Chen et al.,  Softmatch: Addressing The Quantity-Quality Trade-Off In Semi-Supervised Learning, ICLR, 2023.

---

### Official Review · Reviewer_koUF · 2025-03-07

**Overall Recommendation:** 3

**Summary:**

This paper introduces a principled framework for improving pseudolabeling-based semi-supervised learning (SSL) by explicitly controlling confidence scores and thresholds to manage pseudolabel quality and quantity. The approach addresses limitations of heuristic-driven methods, offering a systematic way to balance pseudolabel accuracy and coverage. Extensive experiments including ablation studies confirm the effectiveness of the optimization framework and threshold estimation.

**Claims And Evidence:**

Yes.

**Essential References Not Discussed:**

No.

**Experimental Designs Or Analyses:**

Yes.

**Methods And Evaluation Criteria:**

Yes.

**Other Comments Or Suggestions:**

Please refer to the weaknesses above.

**Other Strengths And Weaknesses:**

Strengths:
1. The paper proposes a approach to learn confidence scores and thresholds via an optimization problem that maximizes pseudolabel coverage while ensuring error rates remain below a target tolerance (ϵ), which addresses limitations of heuristic-driven methods, offering a systematic way to balance pseudolabel accuracy and coverage.
2. The framework could be flexibility combined with popular SSL methods (e.g., Fixmatch, Freematch), enhancing their performance by leveraging high-quality pseudolabels.
3. Ablation studies confirm the effectiveness of the optimization framework and threshold estimation.

Weaknesses:
1. My main concern is that the core technique used in this paper, which estimates thresholds on the test set to filter pseudolabels, closely resembles that of [1], despite the paper citing [1] and acknowledging this in Line 231 “Similar procedures have been used in the context of creating reliable datasets and are backed by theoretical guarantees for the quality of pseudolabels produced”. Since the auto-labeling method proposed in [1] can also be interpreted as a form of pseudolabeling, the contributions of this work appear incremental. The paper should further discuss the differences in insights and techniques compared to [1], such as in their objectives, algorithms, or validation strategies.
[1] Vishwakarma, H., Lin, H., Sala, F., and Vinayak, R. K. Promises and pitfalls of threshold-based auto-labeling. In Thirty-seventh Conference on Neural Information Processing Systems, 2023.

**Questions For Authors:**

None.

**Relation To Broader Scientific Literature:**

The key contributions of the paper are closely related to the fields of semi-supervised learning and pseudolabeling. To leverage unlabeled data to train the predictive model under semi-supervised learning, the authors proposes a approach to learn confidence scores and thresholds via an optimization problem in pseudolabeling unlabeled data that maximizes pseudolabel coverage while ensuring error rates remain below a target tolerance.

**Theoretical Claims:**

Not applicable, as this paper does not put forward any theoretical claims or provide corresponding proofs.

---

> ### Author Rebuttal · Authors · 2025-04-01
>
> Thanks for the careful review and positive feedback.  We appreciate the recognition of the strengths of our work — *a flexible and principled approach for learning confidence scores and thresholds for pseudolabeling and its empirical effectiveness*. Our response to the queries is as follows,
>
> **Differences in insights and techniques compared to [1]**
>
> The prior work [1] studies a procedure to create labeled datasets with accuracy guarantees. Thus, [1] is only concerned with labeling the data correctly and not with how good the end model is. In contrast, our work is on semi-supervised learning, where **our goal is to learn a classifier with good generalization error**. In this procedure, the pseudolabels are never "committed", i.e., in each iteration, the assigned pseudolabel to a point can change. In contrast, in [1], the label assigned to a point is never changed. Furthermore, our approach and insights are novel within the area of pseudolabeling-based semi-supervised learning (SSL) [4,5]. We provide two points on this:
>
> First, our work settles the question of the right choices of confidence functions for SSL. Recent works [2,3] have highlighted the problem of miscalibrated confidence functions, leading to inefficiencies in pseudolabeling-based SSL. Although these works offer solutions to this problem, they still fall short of addressing the core issue: the confidence function is not specifically tailored to the needs of SSL. To provide a principled solution to this problem, we adapt the framework for learning confidence functions in the TBAL setting from [1] to the SSL setting.
>
> Second, we show how this framework for learning confidence functions can work in concert with popular SSL methods such as Fixmatch [4], Freematch [5], etc., and conduct an extensive empirical evaluation demonstrating that using confidence function learned from our method can yield significant improvements in the test accuracy. The flexibility to be integrated into a variety of techniques is novel to this work; works like [1] did not need to offer it. As a result, our work provides a flexible solution to practitioners using SSL, freeing them from the trial and error of selecting various confidence functions or hand-crafting one.
>
> [1] Vishwakarma, H., Lin, H., Sala, F., and Vinayak, R. K. Promises and pitfalls of threshold-based auto-labeling. NeurIPS, 2023
>
> [2] Loh et al., Mitigating confirmation bias in semi-supervised learning via efficient bayesian model averaging. TMLR, 2023.
>
> [3] Mishra et al., Do not trust what you trust: Miscalibration in semi-supervised learning, arXiv, 2024.
>
> [4] Sohn et al., Fixmatch: Simplifying semi-supervised learning with consistency and confidence, NeurIPS, 2020.
>
> [5] Wang et al., Freematch: Self-adaptive thresholding for semi-supervised learning, ICLR, 2023.
>
>
> We hope our response resolves your queries. We are happy to answer any further questions.

---

### Official Review · Reviewer_S5pP · 2025-03-13

**Overall Recommendation:** 3

**Summary:**

This paper proposes PabLo, a novel method for semi-supervised learning. The authors conceive their approach through noting that the threshold for selecting pseudolabels from the teacher model should be both permissive enough to allow for a large degree of supervision, while not being so permissive as to introduce low-quality labels. PabLo proposes solving a surrogate optimization problem during each SSL training iteration, in which pseudolabeling thresholds are chosen to maximize pseudolabel coverage while constrained by a maximum error bound. Additionally, PabLo introduces a pseudolabel accumulation step, in which pseudolabels from previous training iterations can be brought into the current update if it is otherwise missed by the current teacher model. The authors validate their approach on three different image classification datasets, demonstrating an improvement when used in conjunction with common SSL approaches.

**Claims And Evidence:**

The claims presented in the paper are all valid.

**Essential References Not Discussed:**

N/A

**Experimental Designs Or Analyses:**

The experimental design is sound.

**Methods And Evaluation Criteria:**

The proposed experimental methods are valid.

**Other Comments Or Suggestions:**

I believe the paper will flow better if the related work section was moved to the beginning of the paper, after the introduction.

**Other Strengths And Weaknesses:**

Strengths:
- The method is lightweight and can easily be integrated with currently existing SSL approaches.
- Improvement over baselines in the provided experiments is convincing.
- The theoretical framework which motivates PabLo is intuitive, and the careful analysis of the pseudolabeling threshold is principled.

Weaknesses:
- The experiments are limited to a single labeled fraction for each dataset; the results would be more convincing if multiple labeled data fractions were evaluated for each dataset.
- The psuedolabel accumulation step, although introduced as a possible improvement, does not meaningfully improve the test accuracy.
- PabLO includes several hyperparameters chosen heuristically (e.g. $N_{cal}$, $N_{th}$, $\epsilon$). This complexity of selecting all of these limits the practical usability of the method.
- The authors introduce two algorithms for selecting thresholds, either selecting class-wise thresholds or global thresholds. However, it is not made clear which is adopted in the final algorithm, and no comparison between the two is included in the ablation studies.

**Questions For Authors:**

- See weaknesses: does the final algorithm use class-wise or global thresholds (e.g. algorithm 1 or 2)?
- Have the authors made plots visualizing the pseudolabel threshold as a function of training iteration (similar to Fig 5)? It would be interesting to see how the value of the threshold evolves during training, as this is subtly different from the pseudolabel coverage percent plotted in Fig 5.
- What is the authors intention of the analysis on pseudolabel accumulation, as the approach ultimately does not improve PabLo's performance? Have the authors investigated more sophisticated strategies for pseudolabel accumulation (e.g. downweighting the contribution of pseudolabels accumulated from past training iterations)?

**Relation To Broader Scientific Literature:**

This paper investigates thresholding in the context of pseudolabeling beyond the level of simply using heuristic techniques, as has been tried previously in work such as FixMatch. The more principled approach of choosing thresholds improves upon these approaches and can be integrated directly with them.

**Theoretical Claims:**

N/A

---

> ### Author Rebuttal · Authors · 2025-04-01
>
> We thank the reviewer for the constructive feedback and the noted strengths — *a lightweight, intuitive, and theoretical framework to learn scores and thresholds that can be integrated with existing SSL approaches to improve their performance*. Our response to the queries is as follows,
>
> **More labeled data.**  We have evaluated the methods on settings with smaller $N_l$. Specifically, using $N_l=$ 40 for Cifar-10 and SVHN and 400 for Cifar-100. **[The results are available here](https://anonymous.4open.science/r/icml-rebuttal-2024-anon-E247/cifar10_cifar100_svhn_low_label.png)**. The results are consistent with claims in the main paper and even more pronounced than those in Table 2.
>
>
> **On pseudolabel accumulation.** For the accumulation procedure our hypothesis is that it may help the high-precision pseudolabeling methods but it may not be useful for methods with noisy pseudolabels. We do not expect it to help baselines and the results in Table 4 are for completion — to show the performance of baselines with accumulation as well. The accumulation method is generally helpful when used in concert with our confidence scores and thresholds that can ensure high-precision pseudolabels. More sophisticated accumulation strategies accounting for the staleness of pseudolabels might be interesting to explore for future works.
>
>
> **On hyperparameters.** While several of the SSL baselines also require choosing hyperparameters, we provide guidance on selecting the hyperparameters $N_{cal}$, $N_{th}$, $\epsilon$. The results in Table 6, suggest setting $N_{cal}, N_{th}$ to as high value as possible will be favorable. For $\epsilon$ a small value ( $\le$ 1%) is favorable when $N_l$ is not too small (i.e. when the initial models are not expected to be too bad); when we expect the initial models to be bad (or have $N_l$ too small), then using slightly higher epsilon around 5% is favorable. We will include a detailed discussion on these recommendations in the paper.
>
>
>
> **Class-wise vs global thresholds.** Class-wise threshold estimation is suitable when there are less number of classes and global threshold estimation is suitable when the number of classes is large. We use class-wise for CIFAR-10 and SVHN settings and global for CIFAR-100 cases. The rationale behind this is, that when the number of classes is large it is hard to have sufficient samples for each class to learn meaningful thresholds. Thus in such settings, it is more useful to learn a global threshold.
>
> **Moving related works section earlier.**  We will move this in the camera-ready version.
>
> **Thresholds over iterations.** Since we learn new scores in each pseudolabeling iteration, the scale of scores is not consistent across the iterations and thus we do not see a pattern in thresholds over iterations. However, Figures 4 and 5, provide insights into how the quantity and quality of pseudolabels evolve over time.
>
> We hope our response resolves the queries. We are happy to answer any further questions you may have.

---

### Official Review · Reviewer_cbpA · 2025-03-14

**Overall Recommendation:** 2

**Summary:**

The paper proposes PabLO, a framework for improving pseudolabeling-based semi-supervised learning (SSL) by learning confidence scores and thresholds with explicit control over pseudolabeling error tolerance. The core idea is to formulate pseudolabeling as an optimization problem that maximizes coverage while bounding error, replacing heuristic thresholding strategies. PabLO integrates with existing SSL methods and introduces pseudolabel accumulation to reuse high-confidence labels. Experiments on CIFAR-10/100 and SVHN demonstrate significant accuracy improvements over baselines.

## update after rebuttal

The author's response has partially addressed my concerns. After reviewing the other reviewers' comments and the author's rebuttal, I believe this paper still requires further improvements. Therefore, I maintain my rating.

**Claims And Evidence:**

Yes.

**Essential References Not Discussed:**

There are many SSL methods that the authors should ideally introduce and compare experimentally, such as SimMatch (CVPR 2022), SoftMatch (ICLR 2023), AdaMatch (ICLR 2022), and so on.

**Experimental Designs Or Analyses:**

1. In the paragraph titled "Adjusted iterations for baselines," the authors limited the number of iterations for the methods. Have the authors considered the convergence of the method? In the CIFAR100 experiment plots, it appears that the methods have not yet reached the convergence iterations.
2. Regarding the failure of the method in the Fixmatch + SVHN experiment, the authors did not provide a detailed explanation for the cause.
3. The impact of the confidence function architecture (2-layer NN) is unexplored.

**Methods And Evaluation Criteria:**

1. The authors' method essentially uses the validation set to identify confidence functions and thresholds that can be transferred to the unlabeled training set. It seems to implicitly assume a strong alignment between the distributions of these two sets. Is there any theoretical proof that guarantees the optimal solution on the validation set can be transferred to the unlabeled training set? Moreover, have the authors considered whether the method would still be effective in cases where the distributions are not aligned?
2. The pseudolabel accumulation method proposed by the authors appears to be immature. In Table 4, it fails to consistently improve the results across different methods, which significantly limits the generalizability of this approach.
3. In the evaluation, the authors used three datasets, among which CIFAR-10 and SVHN are relatively simple. It is recommended that the authors introduce stronger benchmarks. Furthermore, the authors only conducted experiments on two SSL methods (Fixmatch, Freematch), which is too limited to demonstrate the generalizability of their approach. It would be beneficial to include more baselines to enhance the persuasiveness of their method.

**Other Comments Or Suggestions:**

Nothing.

**Other Strengths And Weaknesses:**

PabLO integrates seamlessly with existing SSL frameworks, but training confidence functions and calculating thresholds increase runtime, and the discussion regarding runtime is not sufficiently comprehensive.

**Questions For Authors:**

1. How does the actual pseudolabeling error during training compare to the target ϵ? Could you provide per-iteration error measurements?
2. Why use a 2-layer NN? Have you explored alternatives?

**Relation To Broader Scientific Literature:**

The work builds on SSL and confidence calibration. It extends prior SSL methods by replacing heuristic thresholds with learned, error-bounded ones.

**Theoretical Claims:**

The paper does not provide formal theoretical guarantees.

---

> ### Author Rebuttal · Authors · 2025-04-01
>
> We appreciate the feedback and the noted strengths of our paper. Our work is well-positioned in the literature on SSL and confidence calibration. Our principled methods to learn confidence scores and thresholds with error bounds replace the heuristic-based choices and enhance the prior SSL methods. Our response to the queries is as follows,
>
> **Generalization/transfer to unlabeled data.** Yes, we assume that the validation data and the unlabeled data are independent and identically distributed (i.i.d). Under this assumption, it is easy to show using standard uniform convergence results that the optimal solution on the validation set will transfer (generalize) to the unlabeled set. More specifically, our method uses parts of the validation data:  $N_{cal}$ samples for learning the confidence scores and $N_{th}$ samples for estimating thresholds. Using the uniform convergence results, we can show that the excess pseudolabeling error when transferring the solution to the unalabeled data will be $O(1/{\sqrt{N_{cal}}} + 1/{\sqrt{N_{th}}})$. Thus, as long as  $N_{cal}$ and $N_{th}$ are sufficiently large and the i.i.d assumption is satisfied, the solution will generalize to the unlabeled data.
>
> The setting where the distributions of unlabeled and validation data are not aligned is interesting, and adapting our methods to these settings would be a fruitful direction for future work.
>
>
> **On pseudolabel accumulation.** For the accumulation procedure, our hypothesis is that it may help the high-precision pseudolabeling methods, but it may not be useful for methods with noisy pseudolabels. We do not expect it to help baselines, and the results in Table 4 support this. The accumulation method is generally helpful when used in concert with our confidence scores and thresholds that can ensure high-precision pseudolabels.
>
> **More baselines and datasets.** Our focus is on principled choices for confidence scores and thresholds for pseudolabeling based SSL, and we proposed solutions to this end. The goal of our experiments is to show that modern and commonly used methods, such as Fixmatch, Freematch, can be adapted to use our scores and thresholds, and the performance of the resulting method is better in comparison to using their standard versions. Since our focus is on confidence functions, we included baselines Margin Regularization (MR) and Bayesian Model Averaging (BAM) that are aimed at improving the calibration of confidence scores for pseudolabeling.
>
> We used common benchmark datasets in SSL, that are sufficient for our empirical analysis. Introducing stronger benchmarks for SSL would be interesting future work.
>
>
> **Fixmatch + SVHN Case.** We point the reviewer to Figure 2 (top, right), where we plot the test accuracy over iterations for this case. This plot suggests that all the methods for the Fixmatch + SVHN case have similar performance.
>
>
> **Related works.** We have updated the related works with a discussion on the shared references.
>
> **Discussion on runtime.** It is correct that additional training for confidence functions and calculating thresholds increases runtime. For this reason, we adjust the training iterations of the baselines so that all the methods are run for the same amount of time. We have included the details on runtime in Appendix C. We are happy to provide any specific details the reviewer is interested in.
>
> On running the CIFAR-100 experiment longer. For fair comparison, we fixed our experiment protocol, i.e., run the variations with our method for 25K iterations and the baselines for the equivalent number of iterations in the same amount of time. Thus, the evaluation is fair. Prior works run till 1 million iterations to reach near convergence. Note that this takes an enormous amount of time, and one of the advantages of using our method is that it can achieve high accuracy earlier and does not require running very long.
>
> **Observed pseudolabeling errors.** We plot the pseudolabeling error and coverage for the Cifar-10 setting. **[Please see the plot on this link](https://tinyurl.com/j2hb77fs)**.  We see that the observed pseudolabeling error is very close to the target $\epsilon$ when it starts to give non-trivial pseudolabeling coverage, and it approaches $\epsilon$ as the training progresses.
>
>
> **Impact of confidence function architecture and its alternatives.** Our framework to learn confidence scores is flexible to work with any choice of $\mathcal{G}$. While the choice of the function class $\mathcal{G}$ is up to the user, in general, it makes sense to use a flexible non-linear function class. A multi-layer neural network with any activation function could be a good fit here. We chose a 2-layer NN since the classification model is doing the heavy lifting of learning the features, so for $g$ we do not need a highly complex network.
>
>
> We hope our response resolves the queries. We are happy to answer any more questions you may have.

---

### Decision · Program_Chairs · 2025-05-01

**Decision:**

Accept (poster)

**Comment:**

The authors propose a new method to optimise thresholds for pseudo-label techniques in semi-supervised learning. The idea is to maximise pseudo-label coverage while maintaining a good accuracy.

The method is sensible, the paper is nicely narrated, and the experimental results are very good (the method can be added as an additional brick to most popular semi-supervised algorithms).

The authors added nice experiments during the discussion phase, and clarified some points. While none of them was really willing to champion the paper, I still recommend acceptance. I strongly encourage the authors to add the additional experimental results to their paper, and also leverage some of the points that were discussed with reviewers. In particular, the use of validation data should be discussed more.